



# Comparison of co–located rBC and EC mass concentration measurements during field campaigns at several European sites

Rosaria E. Pileci[1], Robin L. Modini[1], Michele Bertò[1], Jinfeng Yuan[1], Joel C. Corbin[1,*], Angela Marinoni[2], Bas J. Henzing[3], Marcel M. Moerman[3], Jean P. Putaud[4], Gerald Spindler[5], Birgit Wehner[5], Thomas Müller[5], Thomas Tuch[5], Arianna Trentini[6], Marco Zanatta[7], Urs Baltensperger[1] and Martin Gysel–Beer[1]

[1] Laboratory of Atmospheric Chemistry, Paul Scherrer Institute (PSI), 5232 Villigen PSI, Switzerland
[2] CNR–ISAC—Italian National Research Council, Institute of Atmospheric Science and Climate, via Gobetti 101, 40129 Bologna, Italy
[3] Netherlands Organisation for Applied Scientific Research (TNO), Princetonlaan 6, 3584 Utrecht, the Netherlands
[4] European Commission, Joint Research Centre (JRC), Ispra, Italy
[5] Leibniz Institute for Tropospheric Research (TROPOS), Permoserstrasse 15, 04318, Leipzig, Germany
[6] Regional Agency for Prevention, Environment and Energy, Emilia–Romagna, 40122, Bologna, Italy
[7] Alfred Wegener Institute, Helmholtz Centre for Polar and Marine Research, Bremerhaven, Germany
[*] now at: Metrology Research Centre, National Research Council Canada, 1200 Montreal Road, Ottawa K1A 0R6, Canada

*Correspondence to*: R. L. Modini (robin.modini@psi.ch) and M. Gysel–Beer (martin.gysel@psi.ch)

**Abstract.** The mass concentration of black carbon (BC) particles in the atmosphere has traditionally been quantified with two methods: as elemental carbon (EC) concentrations measured by thermal–optical analysis and as equivalent black carbon (eBC) concentrations when BC mass is derived from particle light absorption coefficient measurements. Over the last decade, ambient measurements of refractory black carbon (rBC) mass concentrations based on laser–induced incandescence (LII) have become more common, mostly due to the development of the Single–Particle Soot Photometer (SP2) instrument. In this work, EC and rBC mass concentration measurements from field campaigns across several background European sites (Paris, Bologna, Cabauw and Melpitz) have been collated and examined to identify the similarities and differences between BC mass concentrations measured by the two techniques. All EC concentration measurements in $PM_{2.5}$ were performed with the EUSAAR–2 thermal–optical protocol. All rBC concentration measurements were performed with SP2s calibrated with the same calibration material as recommended in the literature. The median ratio between observed rBC and EC mass concentrations was 0.92, when considering all data points from all five campaigns, and the corresponding geometric standard deviation (GSD) was 1.5. The minimal and maximal observed values of median rBC to EC mass concentration ratios on single campaign level were 0.53 and 1.29, respectively. This shows that substantial systematic bias between these two quantities occurred during some campaigns, which also contributes to the large overall GSD. On single campaign level, the relative spread of individual rBC to EC mass concentration ratios was typically between a factor of 1.2 and 1.3 (1 GSD), which indicates fairly good precision of both methods.

Despite considerable variability of BC properties and sources across the whole data set, it was not possible to clearly assign reasons for discrepancies to one or the other method, both known to have their own specific limitations and uncertainties. However, differences in the particle size range covered by these two methods were identified as one likely reason for discrepancies. In particular, rBC to EC mass concentration ratios were found to be systematically less than unity, despite applying a correction for small BC cores that remain undetected by the SP2. This was observed when the rBC mass size distribution was shifted towards smaller modal diameter, which occurred during traffic emission dominated episodes.

Overall, the high correlation between rBC and EC mass concentrations indicates that both methods essentially quantify the same property of atmospheric aerosols, whereas systematic differences in measured absolute values by up to a factor of 2 can occur. This finding for the level of agreement between two current state–of–the–art techniques has important implications for studies based on BC mass concentration measurements, for example for the interpretation of uncertainties of inferred BC mass absorption coefficient values, which are required for modelling the radiative forcing of BC. Homogeneity between BC mass determination techniques is very important also towards a routine BC mass measurement for air quality or human health regulations.

## 1. Introduction

Light absorbing aerosols exert a positive radiative forcing through direct absorption of solar radiation. Moreover, their heating can change the atmospheric dynamics and thereby, cloud formation and lifetime (Samset et al., 2018). Despite the relatively small mass abundance of black carbon (8–17 %; Putaud et al. 2010), it dominates the aerosol light absorption in the atmosphere


(Bond et al., 2013). Additional, significant contribution comes from brown carbon (Kirchstetter et al., 2004), tar balls (Adachi
et al., 2019) and mineral dust (Sokolik and Toon, 1999).

Black carbon aerosols possess a unique set of properties: they are refractory (Schwarz et al., 2006), strong absorbers of short–
and long–wave radiation (Bond and Bergstrom, 2006), insoluble in water (Fung, 1990) and composed primarily of graphene–
like sp2–bonded carbon (Medalia and Heckman, 1969). The source of black carbon is the incomplete combustion of
hydrocarbon fuels, including fossil– and bio–fuels (Bond et al., 2013). BC mass concentration data from atmospheric
measurements are used in many applications such as validation of model simulations (Grahame et al., 2014; Hodnebrog et al.,
2014) and quantification of the mass absorption coefficient of BC ($MAC_{BC}$). The latter is defined as the ratio of the light
absorption coefficient caused by BC to the BC mass concentration, and is a crucial parameter in modelling the BC radiative
forcing (Matsui et al., 2018). For these reasons, it is important to assess the accuracy and comparability of different BC mass
measurement techniques.

There is neither an SI (International System of Units) traceable reference method nor a suitable standard reference material for
quantifying BC mass (Baumgardner et al., 2012; Petzold et al., 2013). This presents a challenge for the long–term, routine
monitoring of BC mass concentrations in observation networks such as GAW (Global Atmosphere Watch), ACTRIS
(European Research Infrastructure for the observation of Aerosols, Clouds and Trace Gases) and IMPROVE (Interagency
Monitoring of PROtected Visual Environments). The lack of a reference method is due to variability in the microstructure of
BC produced by different combustion sources (Adachi et al., 2010), the difficulty of isolating BC from other particulate matter,
and the lack of direct mass–based methods selective to BC without interferences (Baumgardner et al., 2012).

In practice, the BC mass is defined operationally through methodologies that use distinct physico–chemical and/or optical
properties of BC in order to quantify its mass concentration in aerosols. The following three different techniques are most
commonly applied: filter based thermal–optical evolved gas analysis (Huntzicker et al., 1982; Chow et al., 2007; Cavalli et al.,
2010); laser induced incandescence (LII) (Schraml et al., 2000; Stephens et al., 2003; Schwarz et al., 2006; Michelsen et al.,
2015) and aerosol light absorption based methods (Rosen et al., 1978; Hansen et al., 1984; Arnott et al., 2003; Petzold et al.,
2005). The specific terms used to refer to the mass of BC quantified by each of these three techniques are: elemental carbon
(EC), refractory black carbon (rBC) and equivalent black carbon (eBC), respectively (Petzold et al., 2013). eBC mass
measurements are not further addressed here, as they rely on prior knowledge or assumed values of the $MAC_{BC}$. Such prior
knowledge is not required for thermal optical measurements of EC mass or for LII measurements of rBC mass.

Thermal–optical analysis (TOA) and the LII technique both make use of the high refractoriness of BC to quantify its mass,
although in a different manner. In comparing these two techniques, it is essential to define what is meant by BC. The popular
Bond et al. (2013) definition of BC is, fundamentally, a summary of the properties of highly–graphitized carbon found in soot
particles. There are, however, other forms of light–absorbing carbonaceous particulate matter (PM), with different cross–
sensitivities for TOA and LII. Recently, Corbin et al. (2019) proposed a refined classification of light–absorbing carbonaceous
PM into four classes: soot–BC, char BC, tar brown carbon and soluble brown carbon, and they provided an overview of the
respective physico–chemical properties. This refined classification provides a useful framework in describing the responses of
TOA and LII. For example tar brown carbon, an amorphous form of carbon, is sufficiently refractory to contribute to EC mass,
whereas it is not sufficiently refractory to cause substantial interference in rBC (Corbin and Gysel–Beer, 2019). Any work that
compares BC measurement techniques should therefore consider the types of carbonaceous material present in the sample.

Very few intercomparisons of EC mass and rBC mass are available in the literature, particularly when it comes to ambient
aerosols, despite the fact that both methods are frequently applied these days. This means that the debate on the comparability
of these two quantities is still largely unresolved. Some studies have shown that the two quantities can agree to within a few
percent (Laborde et al., 2012b; Miyakawa et al., 2016; Corbin et al., 2019), while other studies have shown they can
systematically differ by factors of up to 2 to 3 in either direction (e.g. Zhang et al., 2016; Sharma et al., 2017).

In this work, we examined and quantified the level of agreement or disagreement between BC mass concentrations measured
by the thermal–optical analysis and the LII technique. For this purpose, we compared co–located measurements of EC and
rBC mass concentrations from field campaigns performed at several European sites (Bologna, Cabauw, Paris and Melpitz) in
order to sample different aerosol types. Care was taken to harmonize the applied methods: all thermal optical measurements
were performed with the same temperature protocol (EUSAAR–2, European Supersites for Atmospheric Aerosol Research;
Cavalli et al., 2010) and all SP2 calibrations were done using the same calibration material. This first multi–site
intercomparison allows us to more quantitatively assess the extent to which the EC and rBC concentration measurements agree
or disagree with each other. Potential reasons for discrepancies such as different size cuts, calibration uncertainties and various
interferences are discussed.



## 2. Methods

### 2.1 Sampling campaigns: measurements sites and experimental setup

The observations presented here include measurements from five field campaigns at four different sites, three of which are part of the ACTRIS network (Aerosols, Clouds and Trace gases Research InfraStructure; www.actris.eu). Basic information (site and country, station code, coordinates, altitude and year/season) of each field campaign is summarized in Table S1.

The Melpitz research site of TROPOS (Germany; code: MEL; 51° 320' N, 12° 560' E, 87 m a.s.l.) is located in the lowlands of Saxony, 41 km NE of Leipzig, Germany. The nearest village with about 230 inhabitants is 300 m east of the station. The site is representative for the regional background in Central Europe (Spindler et al., 2012, 2013) since it is situated on a flat meadow, surrounded by agricultural land (Spindler et al., 2010). The area is sometimes influenced by long–range transported air masses from source regions in eastern, south eastern and southern Europe which can contain, especially in winter, emissions from coal heating (van Pinxteren et al., 2019). Two separate field campaigns were performed in summer (from 6 May 2015 to 1 July 2015) and winter (from 2 to 23 February 2017). During the two campaigns, the SP2 was placed behind a nafion dryer (model MD700, Perma Pure) with a $PM_{10}$ inlet about 6 m above ground. The $PM_{2.5}$ sampler for the OC/EC samples was placed nearby. The meteorological conditions and aerosol characteristics encountered during the campaigns are described by Altstädter et al. (2018) for the summer campaign and by Yuan et al. (2020) for the winter campaign.

The KNMI (Koninklijk Nederlands Meteorologisch Instituut) Cabauw Experimental Site for Atmospheric Research (Netherlands; code: CBW; 51° 58' N, 4° 55' E, –0.7 m a.s.l.) is located in the background area of Cabauw, 50 km from the North sea. The nearby region is agricultural in an otherwise densely populated area, and surface elevation changes are at most a few meters over 20 km. During the campaign, the SP2 was placed behind a nafion dryer (model MD700, Perma Pure) with a $PM_{10}$ inlet situated 4.5 m above the ground. The $PM_{10}$ sampler, from which filters off–line OC/EC analyses were carried out, did not include a dryer in the sampling line in line with the GAW recommendations (GAW Report–No. 227). The measurements at this site were performed from 13 to 28 September 2016. The meteorological conditions and aerosol characteristics encountered during the campaign are described by Tirpitz et al. (2020).

The Bologna measurements were performed at the main seat of CNR–ISAC (Consiglio Nazionale delle Ricerche – Institute of Atmospheric Sciences and Climate), in Bologna (Italy; code: BOL; 44° 31' N, 11° 20' E; 39 m a.s.l.). The site is classified as urban background and is located in the Po Valley, a European pollution hot spot due to its orography, meteorological conditions and high presence of human activities, resulting in a large number of anthropogenic emission sources (Vecchi et al., 2009; Putaud et al., 2010; Ricciardelli et al., 2017; Bucci et al., 2018). During the campaign, a $PM_{2.5}$ sampler, not equipped with a drier, was situated at the ARPAE Supersito (inside CNR–ISAC area). The SP2 was located inside a fully instrumented mobile van in the CNR parking area, about 50 m away from the ARPAE Supersito. The instruments in the van were connected to two inlet lines situated on the top of the vehicle at a height of 3 m and connected to the main inlet line with an inner diameter of 5 cm; no size cut was performed. The sampled air was dried to below 30 % relative humidity using two custom–built, silica–gel–loaded diffusion driers. The data presented in this paper were collected from 7 to 31 July 2017. The meteorological conditions and the aerosol properties of this campaign are described by Pileci et al. (in preparation).

The SIRTA Atmospheric Research Observatory (France; code: SIR; 48.713° N 2.208° E; 160 m a.s.l.) is situated in Palaiseau, 25 km South of Paris. The station is characterized as suburban background (Haeffelin et al., 2005). This site is influenced both by fresh and aged black carbon mainly originating from the Paris area. It is impacted by road transport emissions all year round and residential wood burning during the winter (Laborde et al., 2013; Petit et al., 2015; Zhang et al., 2018). The SP2 along with many other instruments was installed in an air–conditioned trailer of the SIRTA measurement platform. For the OC/EC measurements, high–volume samplers with a $PM_{2.5}$ cut–off were available in the same area. The measurements were performed from 15 January to 15 February 2010. EC and rBC concentrations during this campaign have previously been published in Laborde et al. (2013).

### 2.2 Thermal optical analysis

#### 2.2.1 Measurement principle, OC/EC split and involved artefacts

In thermal–optical evolved gas analysis (TOA), carbonaceous particles deposited on a filter are thermally desorbed/reacted in order to determine the total carbon mass. This technique further divides the total carbon (TC) into EC and organic carbon (OC) according to the expectation that EC is refractory in an inert atmosphere while OC is not (Chow et al., 1993; Birch and Cary, 1996; Chow et al., 2004). Therefore, TOA provides operationally defined OC and EC mass rather than fundamental quantities. This basic binary split does not acknowledge that neither OC nor EC are well–defined materials. Instead, carbonaceous matter





in aerosols populates the multidimensional space of chemical and physical properties more or less in a continuous manner (Saleh et al., 2018; Corbin et al., submitted to Aerosol Sci. Technol.). Nevertheless, the binary split approach aims at providing an operationally defined EC mass that corresponds to "true" BC mass as defined on a conceptual level by Bond et al. (2013) and Corbin et al. (2019) (see Sect. 1).

Many variants of thermal protocols exist for the thermal–optical analysis of EC mass (Bautista et al., 2015). The results presented in this study are based on the EUSAAR–2 protocol, which was developed by Cavalli et al. (2010). The EUSAAR–2 protocol was specifically optimized for aerosol typically encountered at European background sites e.g. of the ACTRIS network in order to improve the accuracy and homogeneity of these long–term data sets (Cavalli et al., 2016). Furthermore, based on tests at filter loadings the range 0.2 and 62 µgC cm$^{-2}$, this protocol has recently been selected as the European standard thermal protocol to be applied in air quality networks for the measurement of TC (total carbon), OC, and EC in PM$_{2.5}$ (particulate matter) samples (European Committee for Standardisation Ambient air, 2017; EN16909:2017).

The carbonaceous material, deposited on a punch of a quartz–fibre filter, is thermally desorbed through progressive heating; first in an inert atmosphere of pure helium (He) at multiple moderate temperatures (~500–700 °C) (inert mode) and then in an oxidizing atmosphere (98 % He and 2 % $O_2$) at high temperature (~850 °C). The applied duration and the temperature of each step vary between different thermal protocols, as discussed below. The evolving carbon is catalytically converted first to carbon dioxide ($CO_2$) and then to methane ($CH_4$). $CH_4$ is then quantified using a flame ionization detector (FID) and reported as OC (inert mode) and EC (oxidizing mode) mass. The instrument type applied in this study and most commonly used to perform TOA measurements is the OC/EC analyser manufactured by Sunset Laboratory Inc. (Tigard, OR).

Ideally, all OC would desorb in the inert He atmosphere and EC would exclusively burn off in the oxidizing $O_2$ atmosphere (Chow et al., 1993; Birch and Cary, 1996). In practice, a fraction of carbonaceous matter may be more refractory than the applied separation threshold, while not being BC in a strict sense. This would cause a positive bias in measured EC mass. In addition, a fraction of the OC can pyrolyze in the He step to form pyrolytic carbon (PC), which is thermally stable and only desorbs in the $O_2$ step, thereby causing a charring artefact in the mutual quantification of OC and EC. To correct for this latter effect a laser at 658 nm is used in combined thermal–optical analysis to monitor the light transmission through the loaded filter before and during the analysis: PC is strongly light absorbing, thus leading to a decrease of the transmission signal when it forms upon heating in the inert atmosphere. Later, in the oxidizing atmosphere, both PC and EC are released from the filter resulting in an increase of the transmission signal. The time at which the transmission equals again the initial pre–pyrolysis value is used to separate OC and EC, depending on whether the carbon evolved before or after this "split point", respectively. This thermal–optical transmittance (TOT) approach to correct for PC eliminates potential charring artefacts if the PC has the same mass–specific attenuation cross section as the atmospheric native EC (Yang and Yu, 2002), and if no other light–absorbing material evolves from the sample.

The charring correction can also be done using light reflectance (thermo–optical reflectance method; TOR) instead of transmittance. As reported in the review paper by Karanasiou et al. (2015), EC values of atmospheric samples determined using the TOT method are often up to 30–70 % lower than those determined using the TOR method. This happens because the evaporation of non–absorbing particulate matter during heating affects the reflectance to a greater extent than the transmission signal. Furthermore, high loadings of EC result in saturation effects of both optical signals, again to a greater extent for the reflection compared to the transmission method (Chiappini et al., 2014). These two effects result in better reproducibility and accuracy of the TOT based OC/EC split compared to the TOR approach. Therefore, all EC mass values reported in this study are based on the TOT approach.

The above described assumptions on the optical charring correction are only partially fulfilled, typically leaving charring artefacts as a main source of bias even for optically corrected EC mass data (Chow et al., 2004; Subramanian et al., 2006). Pyrolysis depends on many factors, including the amount and type of organic compounds, temperature steps in the analysis and the residence time at each temperature step. This makes the TOA technique sensitive to the aerosol type collected on the filter. Water extraction experiments have shown that water soluble organic carbon (WSOC) compounds are particularly prone to causing charring (Yu et al., 2002; Piazzalunga et al., 2011; Zhang et al., 2012; Giannoni et al., 2016). Samples with a high WSOC content come e.g. from biomass and wood burning (Hitzenberger et al., 2006; Reisinger et al., 2008; Chen et al., 2015). A filter water–washing step prior to TOA can be used to remove WSOC, thereby reducing charring artefacts and improving comparability of different protocols for EC mass measurements (Yu et al., 2002; Piazzalunga et al., 2011). However, filter pre–washing is generally not applied in long–term monitoring TOA measurements for practical reasons (the washing step is time consuming). In these cases, the charring phenomenon can be reduced by adopting a thermal protocol with a sufficiently long residence time at each temperature step in the He atmosphere to allow for maximum OC evolution (Subramanian et al., 2006; Karanasiou et al., 2015).


The OC/EC split can be also biased by EC pre–combustion: EC can thermally evolve in the presence of oxidizing species (Watson et al., 2005; Corbin et al., 2014, 2015), and soluble inorganic compounds (Chow et al., 2001; Yu et al., 2002) and metal salts (Aakko–Saksa et al., 2018) can catalyse EC pre–combustion. If the amount of EC undergoing pre–combustion is significant relative to the amount of PC formed during the analysis, the optical correction (transmittance or reflectance) is not able to account for it and this may cause an underestimation of the EC concentration.

Moreover, brown carbon on filters can affect the laser correction if it was evolving during the OC steps, thereby causing a positive EC artefact. However, brown carbon absorbs much less than EC at the red wavelength ($\lambda = 635$ nm) of the laser used in the thermal–optical instruments, since its absorbance decreases strongly from the blue–UV region of the electromagnetic spectrum towards the red region (Karanasiou et al., 2015), thereby reducing the potential impact of brown carbon interference. Recently, Massabò et al. (2019) developed a modified Sunset Lab Inc. EC/OC analyser to measure the brown carbon content 215 in the sample by adding a second laser diode at $\lambda = 405$ nm.

Further artefacts caused by carbonate carbon are discussed in Wang et al. (2010), Karanasiou et al. (2015) and Querol et al. (2012). However, the thermal–protocol EUSAAR–2 (used in this work and described in Sect. 2.2.2) was developed to have most carbonate carbon evolving as OC. Nevertheless, cases in which carbonate C contributed to "EUSAAR–2 EC" have been reported in previous studies, such that minor positive artefacts in EC cannot be excluded (Karanasiou et al., 2011).

**2.2.2   EUSAAR–2 vs other existing protocols**

Besides EUSAAR–2, the   IMPROVE–A (Interagency Monitoring of PROtected Visual Environments; Chow et al., 1993, 2007) and NIOSH (National Institute for Occupational Safety and Health; Birch and Cary, 1996) thermal protocols are also commonly used for TOA analysis. Various NIOSH–like protocols (NIOSH–5040, NIOSH–840, NIOSH–850, and NIOSH–870) exist that are all modified versions of the Birch and Cary (1996) and Birch et al. (1999) protocols.

Table 1 summarizes the differences between EC measured with EUSAAR–2 and with other protocols reported in previous literature studies. The use of different thermal protocols can result in a wide elemental carbon–to–total carbon variation by up to a factor of five (Cavalli et al., 2010). In general, it has been observed that protocols with a rather low peak temperature in the inert mode like EUSAAR–2 and IMPROVE generally classify more carbon as EC compared to the NIOSH protocol (Karanasiou et al., 2015). The EnCan–Total–900 protocol has much longer retention time at each temperature step compared 230 to the IMPROVE and NIOSH methods (Huang et al., 2006; Chan et al., 2010).

**2.2.3   Variability of EC measurements with the EUSAAR–2 protocol**

Given the artefacts involved in TOA analysis, different instruments can measure different EC concentrations for the same sample, even if the same thermal protocol is used. For this reason, the Joint Research Centre (JRC) European Reference Laboratory for Air Pollution (ERLAP) organizes annual instrumental inter–laboratory comparisons in order to harmonize 235 measurements from different Sunset instruments that employ the EUSAAR–2 protocol, which typically include 15 to 30 participants. The measurement performances are evaluated using several $PM_{2.5}$ quartz fiber filters collected at a regional background site in Italy. Since the true concentrations of EC or TC in these ambient samples are unknown (due to the lack of suitable reference methods or materials), the expected concentrations are chosen ('assigned') as the robust averages (i.e. with outliers removed) of the TC and EC mass concentrations measured by all participants.

The latest intercomparison yielded an EC/TC ratio repeatability (with the same instrument over time) of 3 % to 8 % and a EC/TC ratio reproducibility (amongst different instruments) of 12 % to 17 % (across 21 participants), where the method precision becomes exponentially poorer toward lower TC contents (<10 µgC cm$^{-2}$) and lower EC/TC ratios (<0.07) (EMEP/CCC–Report 1/2018). Table 2 presents EC bias and variability (see Sect. S1 of the SI for further information) for the instruments used in each campaign (based on data from the ERLAP intercomparison campaign that occurred most recently 245 before or after the campaign in question). The SIRTA campaign EC samples were analyzed by the Institute des Géosciences de l'Environnement (IGE, Grenoble), the Cabauw samples were analyzed by the Joint Research Center (JRC, Italy), the Melpitz (summer and winter) samples were analyzed by the Leibniz–Institut für Troposphärenforschung (TROPOS), and the Bologna campaign samples were analyzed by ARPAE. The EC bias and variability of the instrument used for analyzing the Bologna filter samples, which did not participate in a full ERLAP intercomparison, was determined by comparison with the 250 JRC ERLAP reference instrument for nine filter samples from the Bologna campaign. The EC bias found was smaller than 20 % for all applied OC/EC analyzers, which is within the TOA measurement uncertainty. Therefore, we did not correct the EC measurements reported in this work for these biases.





Blank filters were analyzed for all campaigns. The blank value for EC mass was always below detection limit or negligibly small compared to EC mass on loaded filter samples, such that applying a blank correction does not make a difference for the resulting EC mass concentration.

### 2.3 The Single–Particle Soot Photometer (SP2)

#### 2.3.1    Principle of measurement

Laser induced incandescence occurs when a high–intensity laser is used to heat light absorbing and highly refractory particles to high enough temperatures for them to emit considerable grey/blackbody radiation. LII can be used to quantify rBC carbon mass concentration in aerosols by detecting the emitted thermal radiation, which is approximately proportional to rBC mass. There are different instrumental approaches for LII using both pulsed–shot lasers (Michelsen et al., 2015), and continuous–wave lasers, as in the commercially–available Single–Particle Soot Photometer (SP2, Droplet Measurement Technologies, Longmont, CO, USA).

The SP2 quantifies the rBC mass in individual particles (Stephens et al., 2003; Schwarz et al., 2006; Moteki and Kondo, 2007). When aerosol particles enter the instrument, they are directed into the centre of an intra–cavity Nd:YAG laser beam with a wavelength of 1064 nm where they are irradiated. BC particles absorb the laser light, thereby being heated up and losing their non–refractory coatings. Eventually, the residual BC cores reach their sublimation temperature and incandesce. Two photomultiplier tubes equipped with optical bandpass filters capture part of this thermal radiation. The broad band and narrow 270 band detectors measure light between 350 nm and 800 nm and between 630 nm and 800 nm, respectively (Schwarz et al., 2006). Since the thermal radiation emitted by black carbon particles is proportional to the volume (and mass) of BC in the particle (Moteki and Kondo, 2010), this radiation intensity can be converted to rBC mass using an empirical calibration curve.

#### 2.3.2    rBC mass calibration

The relationship between incandescence signal peak amplitude and BC mass depends on the BC type (Moteki and Kondo, 2010; Laborde et al., 2012a), which means the instrument should be calibrated with a material that represents the type of BC one seeks to measure. Unfortunately, many types of BC are found in the atmosphere, such that it is typically not possible to calibrate the SP2 specifically with atmospheric BC. Instead, a fixed calibration using commercial BC materials is commonly applied. Therefore, potential variation in the chemical microstructure of atmospheric BC results in uncertainty in rBC mass 280 measurements.

In this study, two different batches of fullerene soot (Alfa Aesar; stock 40971, lots FS12S011 and W08A039) were used. The former is recommended as calibration material (Baumgardner et al., 2012) since it was shown to be suitable for quantifying BC in diesel engine exhaust (agreement within 10 % for rBC cores ≤ 40 fg; Laborde et al., 2012b). Calibrations using the latter batch agreed with those using the former batch within 5 %. In this work, three different SP2s (PSI, IGE, AWI) were used to 285 acquire the data. This does however not contribute appreciably to uncertainties, since the reproducibility of measured rBC mass size distributions was shown to be ±10 % during a large SP2 intercomparison involving six SP2s from six different research groups (Laborde et al., 2012b). The SP2 used during the Melpitz campaigns was calibrated using an APM to select the calibration particles by mass. For the other campaigns a DMA was used for size selection and the corresponding particle mass was calculated using effective density data reported in (Gysel et al., 2011). The latter approach results in an additional 290 error of about 10 %.

#### 2.3.3    Potential interferences and artefacts

One of the strengths of the SP2 is that the incandescence signal is not perturbed by the presence of non–refractory matter internally or externally mixed with BC (Moteki and Kondo, 2007; Slowik et al., 2007). However, other types of highly 295 refractory and sufficiently light–absorbing (at 1064 nm) material can incandesce in the SP2 laser. Therefore, SP2 measurements can potentially contain interferences from metals, metal oxides (Moteki et al., 2017), volcanic ash and dust (rarely) (Kupiszewski et al., 2016). Fortunately, such materials are usually observed only rarely in atmospheric aerosols in large enough quantities to cause significant SP2 measurement artefacts. Furthermore, if they are present, in some cases their presence can be identified and ignored when calculating rBC mass. Specifically, potential interference can be determined with 300 the use of the spectral bandpass filters,  which permits the determination of the color temperature of incandescence (Moteki et al., 2017). Recently, Sedlacek et al. (2018) found that rBC–free organic particles that absorb light at 1064 nm can char and form rBC under sufficiently high SP2 laser power, resulting in an rBC overestimate. In general, this artefact is only likely to be relevant in biomass burning plumes that contain organic tar balls that can absorb light at 1064 nm (Sedlacek et al., 2018). Marine engines operated with heavy fuel oil can also produce tar particles, but Corbin and Gysel–Beer (2019) found that the


contribution of such particles to rBC mass was negligible. Furthermore, it is possible to distinguish incandescing tar particles from soot BC with SP2 measurements by examining the ratio of scattering–at–incandescence to incandescence signals (Corbin and Gysel–Beer, 2019).

### 2.3.4    SP2 detection efficiency and detection range

The SP2 lower detection limit depends on both physical limitations of the detection technique and instrument parameters
chosen by the operator (Schwarz et al., 2010). With optimal setup, the SP2 can reach unit counting efficiency for rBC mass $m_{\mathrm{rBC}} \approx 0.12$ fg (Schwarz et al., 2010; Laborde et al., 2012a), which corresponds to an rBC mass equivalent diameter of $D_{\mathrm{rBC}} \approx 50$ nm using a void–free BC bulk density of 1800 kg m$^{-3}$ (Moteki and Kondo, 2010). The lower cut–off size for unit counting efficiency can be larger if the SP2 is not optimally setup. Usually the SP2's counting efficiency is robust down to $D_{\mathrm{rBC}} \approx 80$ nm ($m_{\mathrm{rBC}} \approx 0.48$ fg). We only considered the data of particles with BC cores greater than $D_{\mathrm{rBC}} = 80$ nm in this
study, as exact characterization of the cut–off curve was not performed in all campaigns. Note, poor counting efficiency for BC cores with greater mass than this limit has been reported by Gysel et al. (2012). PALAS soot, which is characterized by very small primary sphere size and very low fractal dimension, resulting in relatively enhanced heat loss. However, we are not aware of studies indicating reduced counting efficiency for atmospherically relevant BC particles, which have larger primary spheres and higher fractal dimension, compared to PALAS soot.

The SP2 was operated downstream of inlets with a 50 % cut–off diameter at 10 µm in every campaign except for the Bologna campaign, where no external upper size cut was applied. In addition, the SP2 is unable to quantify rBC mass above a certain limit due to the saturation of the electronics that record the signals. This saturation limit can be varied via detector gains, with typical settings resulting in upper limits of quantification ranging from $D_{\mathrm{rBC}} \approx 500$ nm to around $D_{\mathrm{rBC}} \approx 1$ µm. Consequently, the total BC mass may be underestimated if BC cores greater than the upper limit of quantification contribute substantially to
total BC mass. Recently, Schwarz (2019) evaluated an algorithm for reconstructing the peak incandescence intensity from the truncated incandescence signals of large BC cores. We did not apply this approach as it only allows increasing the upper limit of quantification by around 15 % in terms of $D_{\mathrm{rBC}}$ without introducing substantial uncertainty in the upper limit of quantification.

The SP2 has no lower number concentration detection limit (in the absence of leaks), while particle counting coincidence
imposes an upper concentration limit when multiple BC particles cross the laser beam simultaneously. Coincidence only caused negligibly low bias in measured rBC mass concentrations for the concentration ranges that were encountered in this study.

### 2.3.5    Methods to correct SP2 data for missing mass below LDL

Two approaches are most commonly used to extrapolate the measured rBC mass size distribution and correct for the missing rBC mass (Schwarz et al., 2006; Laborde et al., 2013). In this study, both methods were applied in order to assess the sensitivity to the correction approach (Sect. 3.1); based on this assessment the first of the two methods described below was determined to be preferable. The two methods are based on fitting the measured rBC mass size distribution with a unimodal lognormal function since BC mass size distributions are generally close to lognormally distributed (e.g. Fig. 1; and Schwarz et al., 2006;
Reddington et al., 2013).

A first approach to correcting SP2 rBC measurements for potentially missed mass is based on extrapolation of the measured size distribution below and/or above the SP2 detection limits. We hereafter refer to this as the "extrapolation method". The corrected rBC mass, $m_{\mathrm{rBC,corr}}^{\mathrm{extrap}}$, is obtained as the sum of the measured mass, $m_{\mathrm{rBC,meas}}$, and a correction term, $\Delta m^{\mathrm{extrap}}$:

$$m_{\mathrm{rBC,corr}}^{\mathrm{extrap}} = m_{\mathrm{rBC,meas}} + \Delta m^{\mathrm{extrap}} \ . \tag{1}$$

Here, $\Delta m^{\mathrm{extrap}}$ is obtained by fitting a lognormal function $\frac{\mathrm{d}m_{\mathrm{fit}}}{\mathrm{d}\log D_{\mathrm{rBC}}}$ to the measured rBC mass size distribution and only
considering potentially missed mass below the lower detection limit (LDL) of the SP2. No correction was applied for potentially missed mass above the upper detection limit (UDL) for two reasons explained in Sect. 3.1.1. Using this approach, the correction term simplifies to the integrated mass of the lognormal fit in the size range below the LDL:

$$\Delta m^{\mathrm{extrap}} = \Delta m_{\mathrm{rBC} < LDL} = \int_0^{D_{\mathrm{LDL}}} \frac{\mathrm{d}m_{\mathrm{fit}}}{\mathrm{d}\log D_{\mathrm{rBC}}} (D_{\mathrm{rBC}}) \ \mathrm{d}\log D_{\mathrm{rBC}} \ . \tag{2}$$





A second commonly applied approach, hereafter referred to as the "fit method", is based on the assumption that the true BC mass size distribution in the submicron size range exactly follows a lognormal function. Under this assumption, the corrected rBC mass, $m_{\mathrm{rBC,corr}}^{\mathrm{fit}}$, is chosen as the integrated mass, $m_{\mathrm{fit}}$, of a lognormal fit to the measured rBC mass size distribution (which includes corrections for contributions below the LDL and above the UDL):

$$m_{\mathrm{rBC,corr}}^{\mathrm{fit}} = m_{\mathrm{fit}} = \int_0^{+\infty} \frac{dm_{\mathrm{fit}}}{d\log D_{\mathrm{rBC}}}(D_{\mathrm{rBC}})\, d\log D_{\mathrm{rBC}}\,. \tag{3}$$

The corrected BC mass obtained with this second approach is composed of four terms (Eq. 4), which are visualized in Fig. 1: i) the measured rBC mass, $m_{\mathrm{rBC,meas}}$ (black solid line), ii) the rBC mass below the SP2 detection limit, $\Delta m_{\mathrm{rBC}<LDL}$ (red shading; Eq. 2), iii) the rBC mass above the SP2 detection limit, $\Delta m_{\mathrm{rBC}>UDL}$ (blue shading; Eq. 5), and iv) the residual area between the fit, $m_{\mathrm{fit}}$ and the measured rBC mass integrated in the range from $D_{\mathrm{LDL}}$ to $D_{\mathrm{UDL}}$ (denoted as $\Delta m_{\mathrm{fitresid}}$; purple shading; Eq. 6):

$$m_{\mathrm{rBC,corr}}^{\mathrm{fit}} = \Delta m_{\mathrm{rBC}<LDL} + \Delta m_{\mathrm{rBC}>UDL} + m_{\mathrm{rBC,meas}} + \Delta m_{\mathrm{fitresid}}\,, \tag{4}$$

where

$$\Delta m_{\mathrm{rBC}>UDL} = \int_{D_{\mathrm{UDL}}}^{+\infty} \frac{dm_{\mathrm{fit}}}{d\log D_{\mathrm{rBC}}}(D_{\mathrm{rBC}})\, d\log D_{\mathrm{rBC}} \tag{5}$$

and

$$\Delta m_{\mathrm{fitresid}} = \int_{D_{\mathrm{LDL}}}^{D_{\mathrm{UDL}}} \frac{dm_{\mathrm{fit}}}{d\log D_{\mathrm{rBC}}}(D_{\mathrm{rBC}}) - \frac{dm_{\mathrm{meas}}}{d\log D_{\mathrm{rBC}}}(D_{\mathrm{rBC}})\, d\log D_{\mathrm{rBC}}\,. \tag{6}$$

Note that with this definition $\Delta m_{\mathrm{fitresid}}$ has a negative value for the example shown in Fig. 1. The correction term in the case of the fit method is naturally defined as the difference between the corrected and the measured rBC mass:

$$\Delta m^{\mathrm{fit}} = m_{\mathrm{fit}} - m_{\mathrm{rBC,meas}}\,. \tag{7}$$

From Eqs. (4) and (7), one can derive:

$$\Delta m^{\mathrm{fit}} = \Delta m_{\mathrm{rBC}<LDL} + \Delta m_{\mathrm{rBC}>UDL} + \Delta m_{\mathrm{fitresid}}\,. \tag{8}$$

Comparing the missing mass correction terms of the two approaches given in Eqs. (1) and (3) shows that the corrected rBC mass differs by the sum of two physically meaningful quantities, the fit residual and the extrapolated rBC mass above the UDL:

$$m_{\mathrm{rBC,corr}}^{\mathrm{fit}} - m_{\mathrm{rBC,corr}}^{\mathrm{extrap}} = \Delta m_{\mathrm{rBC}>UDL} + \Delta m_{\mathrm{fitresid}}\,. \tag{9}$$

The results of these two approaches are compared and discussed in relation to the different datasets used in this study in Sect. 3.1.1. Outside of Sects. 3.1.1 and 3.1.2, Figs. 1, 2 and S1, and Table 4, this manuscript applies the first method (Eq. 2) to quantify rBC mass.

## 2.4 Auxiliary measurements

### 2.4.1 Aerosol size distribution

In the Melpitz winter and summer campaigns, aerosol number size distributions in the diameter range from 3.8 to 770 nm were measured with a mobility particle size spectrometer (MPSS), custom built by Wiedensohler et al. (2012), which consists of a differential mobility analyzer (DMA) and a condensation particle counter (CPC). The DMA was operated with a sheath air flow of 10 L min$^{-1}$ and the aerosol number size distribution was measured every 20 minutes. During the Cabauw campaign, a modified version of a commercially available scanning mobility particle sizer (TSI SMPS 3034) provided the number size distribution of the aerosol in the diameter range from 10 to 470 nm. No size information is available for the SIRTA and Bologna campaigns.

### 2.4.2 Absorption Ångström exponent (AAE) inferred from Aethalometer data





The Aethalometer measures the light attenuation through a sample filter that is continuously loaded with aerosol (Hansen et al., 1984). The raw attenuation coefficient ($b_{atn}$) is calculated from the rate of attenuation change with time. The relationship between attenuation coefficient and absorption coefficient of the deposited aerosol particles is linear for low attenuation values but saturation occurs when the attenuation values are high (Weingartner et al., 2003). Therefore, the measurements must be corrected for this "loading effect" in order to obtain a corrected attenuation coefficient ($b_{atn,corr}$) (Virkkula et al., 2007; Drinovec et al., 2015). The attenuation coefficient is greater than the absorption coefficient due to multi–scattering effects within the filter matrix, described with a proportionality constant $C$.

From a pair of $b_{atn,corr}$ at two different wavelengths, $\lambda_1$ and $\lambda_2$, it is possible to calculate the absorption Ångström exponent, AAE($\lambda_1, \lambda_2$), a coefficient commonly used to describe the spectral dependence of the aerosol light absorption coefficient (Moosmüller et al., 2009):

$$\text{AAE}(\lambda_1, \lambda_2) = -\frac{\ln(b_{atn,1,corr}(\lambda_1))}{\ln(b_{atn,1,corr}(\lambda_2))}\frac{\ln(\lambda_2)}{\ln(\lambda_1)} \tag{10}$$

Note that since the Aethalometer $C$–value has only a small spectral dependence (Weingartner et al., 2003; Corbin et al., 2018), it is possible to infer the AAE directly from the corrected attenuation coefficient, as is done in this work.

The AAE provides an indication on the sources of BC (Zotter et al., 2017). The light absorption of particles from traffic emissions is dominated by BC, which has an AAE of ~1. By contrast, wood burning emissions contain a mixture of BC and co–emitted brown carbon. Light absorption by brown carbon has a much stronger spectral dependence than BC, such that the mixture has an AAE between ~1 and 3 (Kirchstetter et al., 2004; Corbin et al., 2018). This makes relative apportionment of BC to traffic and wood burning sources based on aerosol AAE possible. However, this simple approach only works in the absence of additional BC sources or light absorbing aerosol components (e.g. from coal combustion).

In this paper the AAE values were calculated with the formula presented in Eq. (10) with $\lambda_1 = 470$ nm and $\lambda_2 = 950$ nm. Aethalometer AE–31 (Magee Scientific) instruments were used during the SIRTA and Cabauw campaigns. These measurements were corrected for the loading effect with the algorithm developed by Weingartner et al. (2003). In the other campaigns, Aethalometer AE–33 (Magee scientific) instruments were used. These data did not need further correction since the algorithm developed by Drinovec et al. (2015), which takes into account the filter loading effect, is incorporated in the instrument. However, the AE33 firmware correction was not working properly during Bologna campaign. Therefore, these data were corrected using the Weingartner et al. (2003) correction.

## 3. Results and discussion

### 3.1 rBC mass potentially missing below the LDL of the SP2

#### 3.1.1 Comparison of two approaches to correct for the truncated rBC mass

In the following, we compare the results from the two different approaches for estimating the missing rBC mass concentration outside the size range covered by the SP2 (see Sect. 2.3.5).

Typically, the measured size distributions only approximately followed a lognormal distribution. We chose to infer and present the missed rBC mass estimate based on fitting across the range from 80 nm to 300 nm. In addition, the sensitivity to the fitted range was assessed. The estimated missed rBC mass below the SP2 LDL increased by up to 11 6 of the measured mass when increasing the lower fit limit from 80 nm to 100 nm, i.e. around the SP2 LDL. This provides evidence that the extrapolation towards the smallest BC cores is not strongly affected by the SP2 counting efficiency performance, which could potentially be degraded in this range. The fit approach used the total area of a lognormal fit to rBC mass size distributions to obtain the correct rBC mass. Sensitivity analyses done with fitting to a manually prescribed upper limit between 200 nm and 400 nm showed that the fit results were insensitive to the choice of this upper limit for valid fits. Therefore, a fixed fit range from 80 nm to 300 nm, which always provided good match between measured and fitted size distribution around the mode of the distribution, will be used in the remainder of this manuscript for missing mass corrections. The validity of the fits was determined by comparing the fitted peak location with the mode of the measured data. Data were only fitted to a manually prescribed upper limit, and the fit results were insensitive to the choice of this upper limit for valid fits, while larger deviations occurred for invalid fits.

Detailed results of the missing mass correction are listed in Table 4. The extrapolation method and the fit method provide comparable results. It can be seen that $\Delta m^{extrap}$ varied in the range 3–25 %, while $\Delta m^{fit}$ varied between 3–21 %. Considerable



variability in missing mass correction between campaigns occurred due to differences in the rBC mass size distribution,
especially differences in the average modal diameters, which is also listed in Table 4. As shown in Eq. (9), the extrapolation
and the fit methods for missing mass correction differ by the sum of the fit residual ($\Delta m_{\text{fitresid}}$) and the extrapolated mass
above the UDL ($\Delta m_{\text{rBC}>UDL}$). The opposite signs and comparable magnitudes of these two terms (Table 4), shown as purple
and blue shadings in Figs. 1 and S1, have partially compensating effects, resulting on average in only 3 % difference between
the two missing mass correction methods.

The systematic difference between measurement and fit for rBC mass equivalent diameters near the UDL of the SP2 (Figs. 1
and S1) could indicate either the presence of a second lognormal mode that is centered at a larger diameter than the main mode,
or an inaccurate extrapolation of the incandescence signal calibration for masses greater than 64 fg ($D_{\text{rBC}}$ = 408 nm). As both
effects make extrapolation of the rBC mass size distribution above the UDL uncertain, we decided to apply the extrapolation
method in this study. As explained in Sect. 2.3.5 (Eqs. 1 and 2), the extrapolation method only uses the fit below the LDL of
the SP2 to estimate missing rBC mass. This ensures a well–defined upper cut–off in terms of rBC core mass for the corrected
rBC mass concentration results. In the following, all reported rBC mass concentrations are corrected with the extrapolation
method (Eq. 2) with fit range chosen from 80 nm to 300 nm, unless otherwise stated.

The missing mass correction results for the Melpitz winter campaign are significantly different from those for the Melpitz
summer and all other campaigns (Fig. 1, Fig. S1 and Table 4). Specifically, the missed mass percentage for the Melpitz winter
campaign is less than 3 %, while it is between 18 % and 24.5 % for the other campaigns. This is due to the fact that the Melpitz
winter rBC core mass size distribution peaks in the middle of the SP2 detection range, with $D_{\text{rBC,mode}}$ = 227.9 nm (Fig. 1 and
Table 4). This is not the case for the other campaigns, where, as shown in Table 4, the average rBC mass size distributions
have their maximum between 118.6 and 142.9 nm (Figs. 1 and S1). This could indicate that in Melpitz during the winter the
BC source was different from that of the other campaigns of this work. Indeed, with a back–trajectory analysis on the same
dataset, Yuan et al. (2020) showed that the period between 5 and 14 February 2017 was characterized by air masses transported
from south–east Europe, where coal is still used as fuel (Spindler et al., 2013). Coal combustion and biomass burning produce
rBC size distributions with larger modal diameter than traffic emissions (Bond et al., 2013; Liu et al., 2014; Schwarz, 2019).

### 3.1.2    Limits to rBC mass missed in small BC cores imposed by the BC particle number

The presence of an additional mode of small particles below the lower detection limit of the SP2 would introduce an error in
the above extrapolation calculations. Indeed, a substantial fraction of nascent soot particles emitted by combustion engines is
usually below the detectable size range of the SP2. Count median diameters (CMD) of non–volatile particle size distributions
in aircraft turbine exhaust range from 15 to 40 nm (Lobo et al., 2015; Durdina et al., 2017, 2019) while unfiltered gasoline
direct injection and Diesel engines have larger CMD values ranging from 50 to 100 nm (Burtscher et al., 2001;
Momenimovahed and Olfert, 2015).

The existence of additional modes of BC cores at diameters below the SP2 lower detection limit has been hypothesized based
on the observation of 'upticks' in rBC mass size distributions at the LDL of the SP2 (i.e. increasing particle concentration with
decreasing mass equivalent diameter as the SP2 LDL is approached) (Liggio et al., 2012; Cappa et al., 2019). Cappa et al.
(2019) performed multi–modal fits to measured SP2 size distributions with upticks assuming fixed modal diameter (47 nm)
and geometric standard deviation (1.63) of the lognormal mode lying below the SP2 LDL. These authors estimated that the
campaign average mass concentration of the hypothesized small mode of BC particles was as large as 52 % of the total
measured rBC mass concentration. While upticks at the lower end of SP2 size distributions may indicate the presence of an
additional mode of small rBC particles, it should be noted that these upticks might also represent measurement artefacts. SP2
measurements of rBC cores with diameters below 100 nm are sensitive to small variations in fitted calibration curves and it is
difficult to perform accurate calibration measurements near the LDL of the SP2 (Laborde et al., 2012a). Nevertheless, we
cannot exclude presence of an undetected mode with small modal diameter between around 40 nm and 60 nm BC core size in
our studies. Even smaller mode diameter is considered unlikely because such small particles can be found only in the proximity
of a source (Zhu et al., 2006). Larger mode diameter is unrealistic in our campaigns because we did not see any sign of the
upper tail of such a hypothetical mode at the bottom end of the BC size distribution measured by the SP2.

The mass of BC particles below the lower detection limit of the SP2 ($D_{\text{rBC}}$ < ~80 nm) can be estimated by measuring the total
number concentration of non–volatile particles by thermo–denuded MPSS measurements, assuming that BC particles dominate
the number of nonvolatile (NV) particles remaining after thermal treatment (Clarke et al., 2004). Miyakawa et al. (2016)
employed this approach to conclude that the fraction of small rBC particles with $D_{\text{rBC}}$ less than around 80 nm did not contribute
substantially to the total rBC mass concentrations measured at an industrial site south of Tokyo, Japan. In the absence of such
thermally–treated measurements, we assume 30 % of total measured particle number concentration as an upper limit for total



BC particle number concentration (Wehner et al., 2004; Reddington et al., 2013; Cheung et al., 2016). This provides, after subtraction of the BC particle number concentration measured by the SP2, an upper limit ($n_{\mathrm{limit}}$) for the undetected BC particle number concentration.

The three quantities BC particle number concentration, rBC mass concentration and rBC mass equivalent diameter are unambiguously related for a hypothetical perfectly monodisperse mode of BC particles. This relationship is illustrated in Fig. 2, which presents BC number concentration versus rBC mass equivalent diameter along with isolines of constant rBC mass concentration (dash dotted lines). For example, the purple marker indicates that a BC particle number concentration of 494 cm$^{-3}$ and rBC core diameter of 40 nm translates to an rBC mass concentration of 0.03 µg m$^{-3}$. The horizontal dashed lines in Fig. 2 indicate the estimated upper limit, $n_{\mathrm{limit}}$, for BC particle number undetected by the SP2. Taking the Melpitz winter campaign

as an example, the dashed blue line is clearly below the oblique continuous blue line in the BC core range between 40 nm to 60 nm. Comparing the rBC mass concentrations corresponding to these two lines at 40 nm and 60 nm diameter shows that the maximal undetected rBC mass concentration associated with small BC cores is at most 7 % to 23 % of the measured rBC mass concentration for modes peaking within these size limits. For Cabauw, the number limit and mass concentration lines cross at 55.5 nm (green point). The intersect implies that an undetected mode peaking at this size could at most contribute as much

additional rBC mass as measured by the SP2. The constraints resulting for undetected modes between 40 nm to 60 nm are additional 37 % to 125 % of observed rBC mass, respectively. For Melpitz summer, the number limit only provides a very weak constraint on the missed mass as the intercept occurs at 43.5 nm (red point). Therefore, the undetected rBC mass could reach up to 263 % of detected rBC mass if the modal diameter was located at 60 nm.

Based on the discussion in Sect. 3.1.1, we applied in this study the extrapolation method to correct for estimated rBC mass below the SP2 LDL ($\Delta m^{\mathrm{extrap}}$ in Table 4). The resulting corrections are smaller than the upper limit imposed by BC particle number as discussed here. Hence, it cannot be excluded that the truly missed mass was larger than accounted for. The conservative estimate based on the BC particle number considerations suggests that the missing mass could be as large as 23 % (applied correction: 3 %), 125 % (applied correction: 22 %), 263 % (applied correction: 25 %) for the Melpitz winter, Cabauw and the Melpitz summer campaigns, respectively.

### 3.2 Comparison of observed EC and rBC mass concentrations

Here we aim at a quantitative comparison of rBC (after correction using Eq. 2) and EC mass concentrations measured by the SP2 and the thermal–optical method, respectively. Figure 3 shows a scatter plot of time–resolved data using distinct colours for each campaign (rBC data averaged according to the sampling periods of the EC samples). Figure S2 presents the corresponding statistics of the rBC to EC mass ratio and Table 5 reports all statistical parameters. The median values of the

rBC to EC mass ratio lie between the arithmetic and geometric means, indicating distributions that are between the normal and lognormal distribution (Fig. S2). For this reason, we adopted the median when reporting the ratio of the two quantities and in the figures (lines in Figs. 3 and 6), and the geometric standard deviation (GSD) to report the $m_{\mathrm{rBC}}/m_{\mathrm{EC}}$ variability (Table 5). Considering all data points from all campaigns, the median value of $m_{\mathrm{rBC}}/m_{\mathrm{EC}}$ was 0.92 with a GSD of 1.5. That is, $m_{\mathrm{rBC}}$ was on average 8 % smaller than $m_{\mathrm{EC}}$, and 68 % of the individual data points fell into the range within a factor of 1.5 around the

geometric mean ratio. Accordingly, the overall statistics for these two quantities agree closely, with geometric mean values (GSD) of 0.41 (2.60) µg m$^{-3}$ and 0.47 (2.46) µg m$^{-3}$ for $m_{\mathrm{rBC}}$ and $m_{\mathrm{EC}}$, respectively, both ranging from 0.05 to 3.22 µg m$^{-3}$ (Fig. 3).

The above result suggests a very small overall systematic bias between rBC and EC mass on average. However, a look at the statistics calculated for each campaign separately (Table 5 and Fig. S2) reveals a slightly different picture: the variability of

the rBC to EC mass ratio is considerably smaller for individual campaigns, with GSDs typically around 1.2–1.3, and the systematic bias on campaign level is substantially greater than the overall bias, with median ratios ranging from 0.53 to 1.29. During the Melpitz winter campaign, $m_{\mathrm{rBC}}$ was on average 29 % higher than m$_{\mathrm{EC}}$ with $m_{\mathrm{rBC}}$ and m$_{\mathrm{EC}}$ geometric means of 1.20 (2.64) µg m$^{-3}$ and 0.97 (2.16) µg m$^{-3}$, respectively. During the Melpitz summer campaign, $m_{\mathrm{rBC}}$ was comparable to $m_{\mathrm{EC}}$ within 3 %, with respective geometric means of 0.17 (1.57) µg m$^{-3}$ and 0.18 (1.54) µg m$^{-3}$. For the Bologna summer

campaign, the median rBC to EC mass ratio was 0.65, with $m_{\mathrm{rBC}}$ and $m_{\mathrm{EC}}$ geometric means of 0.40 (1.46) µg m$^{-3}$ and 0.64 (1.45) µg m$^{-3}$. The largest difference was found in Cabauw, with a median rBC to EC mass ratio of 0.53 and geometric means of $m_{\mathrm{rBC}}$ and $m_{\mathrm{EC}}$ of 0.46 (1.62) µg m$^{-3}$ and 0.86 (1.63) µg m$^{-3}$, respectively. During the SIRTA campaign, $m_{\mathrm{rBC}}$ was 20 % higher than $m_{\mathrm{EC}}$; this value is somewhat higher than the value of 15 % previously published in Laborde et al. (2013), which is explained by the fact that here we used the $m_{\mathrm{rBC}}/m_{\mathrm{EC}}$ median value instead of the result of the linear fit.

### 3.3 Discussion of level of agreement between the rBC and EC mass concentration measurements

In this section, we test different hypotheses for the observed differences between rBC and EC mass.





### 3.3.1 Differences in upper cut–off diameters and in inlet losses

Differences in the upper cut–off diameters for the EC and rBC mass measurements are a potential source of discrepancy. The EC mass measurements presented in Fig. 3 relate to an upper 50 % cut–off at an aerodynamic particle diameter $D_{aero} = 2.5$ µm at ambient RH (Table 2), except for Cabauw, where a PM$_{10}$ inlet was used. The SP2 measurements were mostly taken behind PM$_{10}$ inlets. However, the SP2 has a more stringent intrinsic UDL, which varied from $D_{UDL} = 439$ nm to 766 nm BC core mass equivalent diameter, depending on the campaign (Table 3). To explore the possibility that BC particles with diameters between the UDL of the SP2 and 2.5 µm aerodynamic diameter contributed to the discrepancies between $m_{rBC}$ and $m_{EC}$, the SP2–related mass equivalent diameters ($D_{ve}$) were converted to aerodynamic diameters ($D_{aero}$). This was done by numerically solving Eq. (11), where $C_C$ is the Cunningham slip correction factor, $\rho_p$ the particle density, $\rho_0 = 1000$ kg m$^{-3}$ and $\chi$ is the particle dynamic shape factor (more details in Sects. S2 and S3 of the SI):

$$D_{ve} = D_{aero} \sqrt{\frac{\rho_o \, \chi \, C_C(D_{aero})}{\rho_p C_C(D_{ve})}}. \tag{11}$$

During the Melpitz winter campaign, the intrinsic UDL was at $D_{rBC} = 722$ nm. The aerodynamic diameter of externally mixed bare BC cores of this size varies from around $D_{aero} = 625$ nm for fractal–like shape to 970 nm for compact shape (Table S3). For coated BC particles, the corresponding dry aerodynamic diameter ranges from around 1140 nm to 1660 nm for coating to core mass ratios of 1:1 and 6:1, respectively. The actual BC mixing state was measured by Yuan et al. (2020), though at smaller core diameters. Using these data as a constraint provides around 1320 nm as a best estimate for the dry aerodynamic diameter. However, the impactor for the filter sampling is operated at ambient RH, which means that hygroscopic growth affects the cut–off diameter. Potential hygroscopic growth was assessed as described in Sect. S3. Accordingly, the aerodynamic diameter of particles with BC cores size at the SP2 UDL increases up to 1610 nm and 2230 nm at 80 % and 95 % RH, respectively, for the best estimate BC mixing state. Externally mixed bare BC particles are not affected by hygroscopic growth. Based on this analysis, it can be expected that the intrinsic SP2 UDL translates to a cut–off varying between PM$_1$ and PM$_{2.5}$, or even slightly smaller or greater under extreme assumptions. This statement also applies for the Melpitz summer campaign, where the SP2 UDL differed only marginally from that of the Melpitz winter campaign (Table 3).

Since during the Melpitz campaigns the EC mass concentrations were measured behind both PM$_1$ and PM$_{2.5}$ inlets, we were able to calculate the fraction of EC mass in particles with aerodynamic diameters between 1 and 2.5 µm out of the total EC mass in PM$_{2.5}$. Fig. 4a and 4b indicate that between 10 and 60 % of EC PM$_{2.5}$ mass was present in the large size fraction (1–2.5 µm) for the majority of measurements during both the Melpitz winter and summer. These coarse EC fractions are greater than the longer–term average values at the Melpitz site, which is potentially related to the fact that coal combustion was a likely source of coarse mode EC at the Melpitz site, at least during the winter campaign (van Pinxteren et al., 2019; Yuan et al., 2020).

The facts that the coarse fraction contributed on average around 30 to 40 % EC mass in PM$_{2.5}$ during the Melpitz campaigns, and that the SP2 BC particle cut–off is likely between PM$_1$ and PM$_{2.5}$, makes it possible that upper cut–off related differences contribute to the discrepancies between measured rBC and EC mass seen in Fig. 3. However, additional covariance analyses of the coarse EC fraction with the rBC to EC bias did not provide a conclusive result. Furthermore, such cut–off effects should rather result in an rBC mass being lower than the EC mass, opposite to the result for the Melpitz winter campaign. This indicates the presence of other effects/biases, which over–compensated for the coarse particle mass that the SP2 was not able to detect.

Concerning Cabauw, coarse mode EC could be a potential cause of the observed low rBC mass to EC mass ratio, given that EC was measured behind a PM$_{10}$ inlet and that the SP2 cut–off was at $D_{rBC} = 537$ nm, resulting in a wider upper cut–off gap than during the other campaigns. However, the rBC modal diameter measured by the SP2 was the second lowest of all campaigns (Table 4 and Fig. S1), which makes a potential bias originating from the lower end of the BC size distribution more likely. A closer assessment is however not possible as no PM1 EC samples are available, which also applies for the other sites.

Differences between $m_{rBC}$ and $m_{EC}$ can also come from differences in the inlet line losses. Particle losses can be caused by the presence of a dryer in the inlet line to which the SP2 was connected. In this work, the dryer losses are estimated to be less than 10 % (see further details in Sect. S4). Although this is, therefore, not the major contributor to the observed discrepancies, it should be addressed in future campaigns.

### 3.3.2 Filter loading and EC/TC ratios





Filter overloading with EC can interfere with the optical detection of pyrolytic carbon, potentially leading to a systematically
low bias in the reported EC mass concentrations. For aerosol collected at an urban location, Ram et al. (2010) reported that
linearity between transmission and EC surface loading was maintained when EC surface loading was kept below 8.0 µg cm$^{-2}$.
Figure S3a presents the observed rBC to EC mass ratios as a function of EC surface loading. Several samples collected during
the Melpitz winter campaign exceeded the above loading threshold (red shading). However, the rBC to EC mass ratio of these
data points was very similar to the other filter samples of the Melpitz winter campaigns with lower surface loading. Moreover,
no systematic trend exists between surface loading and rBC to EC mass ratio for all other campaigns, where EC surface loading
anyway stayed below the above threshold (Table S2). Instead, the bias depends systematically on the campaign as already
shown above. Consequently, filter overloading cannot explain the $m_{\text{rBC}}-m_{\text{EC\_PM2.5}}$ discrepancy during the Melpitz winter
campaign, or for the other campaigns of this study.

The precision of thermal–optical EC mass measurements has been found to degrade at TC surface loadings <10 µg cm$^{-2}$ and
at low EC to TC mass ratio (Sect. 2.2.3). Figure S3b presents the observed rBC to EC mass ratios as a function of TC surface
loading, which frequently dropped to low values between 2 and 10 µg cm$^{-2}$ during 3 out of the 5 campaigns (Table S2).
However, this analysis does not suggest increased random noise nor systematic bias caused by low TC surface loading (points
within red shaded area). Instead, systematic campaign–dependent bias dominates again.

OC/EC split related artefacts in thermal–optical EC mass are more likely to occur at low EC/TC mass ratios. Figure S3c
presents the observed rBC to EC mass ratios as a function of EC/TC mass ratio. No systematic dependence on EC/TC was
found, except possibly for the Melpitz winter campaign. However, multiple other aerosol properties exhibited covariance with
EC/TC on a campaign–to–campaign basis, as will be addressed in Sect. 3.4. Causality hence remains elusive.

### 3.3.3 Systematic EC and rBC bias due to the presence of particular types of particulate matter such as brown carbon or biomass burning BC

As discussed in Sect. 2.3.2, the SP2 sensitivity depends on the BC type. Therefore, differences in the BC properties between
the atmospheric rBC samples and the calibration material may result in systematic bias. The AAE of an aerosol provides
information on brown carbon co–emitted with BC and through this on potential BC sources (Sect. 2.4.2). Figure 5 presents the
relation of $m_{\text{EC}}$ and $m_{\text{rBC}}$, color–coded by the AAE to investigate a possible influence by the presence of brown carbon. Three
zones can be distinguished (see also Table S4 for AAE statistics): the upper part of the figure, with $m_{\text{BC}} > 0.3$ µg m$^{-3}$ and
$m_{\text{rBC}} > m_{\text{EC}}$, represents data collected during the winter campaigns of Melpitz and SIRTA, with an AAE above average (>
1.2; blue symbols) and geometric mean AAE values of 1.36 and 1.38, respectively. The lower right part of the figure, with
$m_{\text{BC}} > 0.3$ µg m$^{-3}$ and $m_{\text{rBC}} < m_{\text{EC}}$, represents data from Bologna and Cabauw, with $0.8 < \text{AAE} < 1.2$ and geometric mean
AAE values of 1.04 (red symbols). The data in the lower left part of the graph, with $m_{\text{BC}} < 0.3$ µg m$^{-3}$, represent Melpitz
summer data, with AAEs between 0.93 and 1.28 (up triangle markers). While there is a general increase in the relative
difference between $m_{\text{EC}}-m_{\text{rBC}}$ with increasing AAE when considering all campaigns (Fig. S4), it is not explained with the
AAE variability within an individual campaign (marked with different colours). We conclude that the variation of BC sources
implied by AAE variability may contribute to variations in the discrepancy between $m_{\text{EC}}$ and $m_{\text{rBC}}$, while not being the main
driver of it.

### 3.4 Reconciliation of sources of discrepancy between rBC and EC mass

The results presented in Fig. 3 and discussed in Sect. 3.2 showed agreement within 8 % between rBC and EC mass
concentrations when averaging over all data points from all campaigns. High correlations were found for individual campaigns,
however, with large variability of the campaign median rBC to EC mass ratios, ranging from 0.53 to 1.29. The analyses
presented in Sect. 3.3.3 (Figs. 5 and S4) suggest some relationship between observed discrepancy and BC source type.
However, many aerosol properties related to potential artefacts are cross–correlated, which makes it difficult to identify causal
reasons.

The lowest rBC to EC mass ratios of 0.53 and 0.65 were observed during the Bologna and Cabauw campaigns (Table 5). The
sampled aerosol during these two campaigns was characterized by smallest BC core sizes, highest EC/TC ratios and lowest
AAE of ~1. The latter shows that BC was dominated by traffic sources. This is the type of ambient aerosol, for which the EC
mass measurement should be quite reliable (e.g. Khan et al., 2012). The common calibration approach of the SP2 (see
Sect. 2.3.2) should only cause limited bias in this case, as it was tailored to match the instrument sensitivity of rBC. Coarse
BC particles with sizes between the upper cut–off diameter of the SP2 and the PM inlet cut–off diameter of the EC sampling
may contribute to the rBC mass being lower than EC mass, though the analyses presented in Sect. 3.3.1 did not provide a clear
result on the importance of this effect. At the lower end of the BC mass size distribution, rBC mass data were corrected for the
missed rBC mass associated with small BC cores (Sects. 2.3.5 and 3.1.1). However, this correction would not account for an



additional BC mode below the SP2 cut–off, as e.g. hypothesized by Liggio et al. (2012) and Cappa et al. (2019), nor could presence of such a mode be excluded by means of particle number–based considerations.

The results for the Melpitz winter campaign are different in many aspects: highest average rBC to EC mass ratio of all campaigns (1.29), largest BC core sizes, highest EC filter loading, and highest AAE (Tables S2 and S4). Large artefacts from missed rBC mass below the SP2 LDL could be excluded with number–based considerations (Sect. 3.1.2). The aerosol
contained a substantial fraction of coarse BC, based on parallel EC measurements made with $PM_1$ and $PM_{2.5}$ inlet cut–off diameters. The SP2 might have missed some of these coarse particles. However, this effect likely caused less than 20 % negative bias in rBC mass concentration, which would have made the discrepancy between rBC and EC mass greater rather than smaller. Based on previous studies (van Pinxteren et al., 2019) and measured AAE, the BC contained substantial contributions from coal burning and/or wood burning emissions. For wood burning BC, this could result in rBC mass that is
low by less than ~20 % due to potentially lower sensitivity of the SP2 (Laborde et al., 2012a). The sensitivity of the SP2 to BC from coal burning is unknown, but the bias is expected to be <30 %. As for the EC, the analyses presented in Sects. 3.3.2 and 3.3.3 did not provide evidence of a clear bias of the EC mass measurements in one or the other direction.

An agreement within 15 % between rBC and EC mass was observed for the Melpitz summer and SIRTA campaigns (Table 5), where the pertinent aerosol and BC properties assumed mean values compared to the range covered by data from all campaigns.
This finding does not exclude compensating errors in one or both measurements. However, no clear evidence for such errors was observed.

### 3.5 Comparison with previous rBC and EC intercomparison studies

In this section, we put our results into context with previous rBC and EC mass intercomparison studies available in the literature. Figure 6 contains a compilation of co–located measurements presented as a scatter plot. The data collected in this
study are shown with green points, the corresponding median ratio is shown by the green line, and the green area illustrates the 1 GSD range around the geometric mean value. In the same graph, data from previously published ambient, lab and chamber studies are reported, including labels indicating the thermal–optical protocol used for the EC measurements. Further information on the data shown Fig. 6 is given in Table S5 (SP2 calibration material, $m_{EC}$ cut–off, TOA thermal protocol, period, location, site characteristics or aerosol source, and average result of the intercomparison).

The data points of the previous studies are scattered around the 1:1 line and the majority of them lie within the 1 GSD range of this study. Therefore, the previous studies confirm the finding that the TOA and the SP2 techniques both provide a consistent measurement of BC mass within the uncertainties of either technique. More specifically, the chamber experiments with CAST soot by Laborde et al. (2012b) show agreement between rBC and EC mass within 15 % (topmost cyan point in Fig. 6). Such close agreement is not surprising as the sample comprised almost pure BC, which simplifies the EC mass measurement, and
the BC mass size distribution was almost completely within the range covered by the SP2. Corbin et al. (2019) investigated exhaust from a four–stroke ship diesel engine (brown triangles in Fig. 6). Close agreement within a few percent was achieved under engine operation conditions under which the emitted refractory carbon was dominated by soot–BC. By contrast, rBC to EC mass ratios substantially smaller than unity were observed when operating the engine under conditions leading to a high fraction of tar brown carbon in the exhaust. This discrepancy could be attributed to a positive interference in EC mass caused
by tar balls. Miyakawa et al. (2016) measured ambient aerosol at an urban location and found very high correlation and close agreement (within 7 %) between rBC and EC mass (after applying line loss corrections, since they found the particle transmission efficiency of the diffusion dryer of the SP2 line to be 84 %). Zhang et al. (2016) reported an average rBC to EC mass ratio of 0.72 for an urban background site, with all data points highly correlated and therefore within the green shading. During the campaign at a remote Arctic site by Sharma et al. (2017), the rBC mass was found to be a factor of 0.51 lower than
the EC mass, with half of the data points lying outside the green shading. They attributed this large bias to two potential reasons: first, due to filter loadings being around the limit of quantification of the TOA, and second, due to large charring bias, causing EC mass overestimation despite optical correction. The mean EC mass uncertainty during the campaign was ~ 28 %, reaching values as high as 80 % during summer due to very low EC mass concentrations.

As discussed in Sect. 2.2.3 and shown in Table 1, the difference in the thermal–optical protocols used to quantify $m_{EC}$, can
result in a bias of ±40 %. For example, the geometric mean ratio between rBC and EC mass of the Zhang et al. (2016) data points would increase from 0.72 to 0.96, if they had been measured with the EUSAAR–2 protocol, or if the 25 % systematic difference between the IMPROVE and EUSAAR–2 protocols as reported by Han et al. (2016) were applied (see Table 1). However, our campaign–to–campaign variability of the rBC to EC mass ratio of roughly ±50 % using the same TOA protocol, (Fig. 3), can be even bigger than the variability associated with a different TOA protocol for the same sample. Therefore, the
rBC mass measured by the SP2 cannot be used to identify the optimal TOA protocol.

### 4. Conclusions



In this work co–located EC and rBC mass concentration measurements from five field campaigns performed in the time period 2010–2017 across several European sites (Paris, Bologna, Cabauw and Melpitz) were collated and examined to identify the differences between BC mass concentrations measured by the thermal–optical analysis and the LII technique. All EC concentration measurements were performed with the EUSAAR–2 thermal protocol, with the TOT technique on quartz filters sampled with high volumes with $PM_{2.5}$ cut–off (except for the Cabauw campaign during which $PM_{10}$ was sampled). All the OC/EC TOA instruments used to perform the EC analysis were compared at the JRC European Reference Laboratory for Air Pollution (ERLAP) to check the EC bias and variability. All rBC mass concentration measurements were performed with SP2s. Three different SP2 instruments (PSI, IGE and AWI) were used in these campaigns, calibrated with the same standard material, fullerene soot, using two different batches, which produced almost identical calibration curves. The mass of BC cores smaller than the lower SP2 detection limit was calculated for all the campaigns including sensitivity analyses. The estimates of missed rBC mass outside the detection range of the SP2 was found to vary between campaigns due to differences in the size distributions of the BC particles.

Overall, considering the five field campaigns, the median of the observed rBC to EC mass ratios for the whole dataset was 0.92, with a GSD of 1.50. The median ratio varied from 0.53 to 1.29 from campaign to campaign. Potential reasons for discrepancies are as follows: source–specific SP2 response, the possible presence of an additional mode of small BC cores below the LDL of the SP2, differences in the upper cut–off of the SP2 and the inlet line for the EC sampling, or various uncertainties and interferences from co–emitted species in the EC mass measurement. The discrepancy between rBC and EC appears to be systematically related to the BC source, i.e. traffic versus wood and/or coal burning. However, it was not possible to identify causalities behind this trend due to potential cross–correlations between several aerosol and BC properties relevant for potential biases. For future intercomparison studies, it is important to constrain the upper cut–off and potential inlet losses of both methods in such a manner that these can be excluded as a source of discrepancy.

The comparison with already published studies showed that most of the rBC to EC mass ratio data points of the other campaigns were within 1 GSD of the median and GSD found in this work. Although in this work, all EC concentrations were measured by the EUSAAR–2 protocol, we note that our reported variability in the rBC to EC mass ratio is greater than the variability expected between EC concentrations measured by different thermal protocols.

From this study, we conclude that the two methods essentially measure the same quantity, i.e. both provide an operationally defined measure of atmospheric BC mass in good overall agreement. However, systematic discrepancies up to ~±50 % were observed at some sites. Lack of a traceable reference method or reference aerosols combined with uncertainties in both of the methods, made it impossible to clearly quantify the sources of discrepancies, or to attribute them to one or the other method.

**Acknowledgment**

The authors are grateful to Achim Grüner for the technical support during the two Melpitz campaigns. We also thank Claudia Zigola (ARPAE) who performed the TOA analysis of the filters during the Bologna campaign. The logistic support of CNR–ISAC technical staff (Francescopiero Calzolari) during the Bologna campaign is appreciated. Many thanks to the technical support of Günther Wehrle during the Bologna campaign.

**Financial support**

This work received financial support from the ERC (grant ERC–CoG–615922–BLACARAT) and from the ACTRIS–2 project (EU H2020–INFRAIA–2014–2015, grant agreement no. 654109; and Swiss State Secretariat for Education, Research and Innovation, contract number 15.0159–1; the opinions expressed and arguments employed herein do not necessarily reflect the official views of the Swiss Government). In addition, the field campaigns were supported by the Transnational Access scheme of ACTRIS–2 project. MZ received financial support from by the Deutsche Forschungsgemeinschaft (DFG, German Research Foundation) – Projektnummer 268020496 – TRR 172, within the Transregional Collaborative Research Center "ArctiC Amplification: Climate Relevant Atmospheric and SurfaCe Processes, and Feedback Mechanisms (AC)3".

**Author contributions**

REP, MB and RLM took the rBC measurements and/or analyzed the raw data during the Bologna campaign. AT was responsible for the EC measurements during the Bologna campaign. AM coordinated the Bologna campaign. JY, MZ and RLM took the measurements and/or analyzed the raw data during the Melpitz winter campaign while JCC took the measurements and analyzed the raw data during the Melpitz summer campaign. TT, TM and BW took the data and coordinated the Melpitz campaigns. GS was responsible for the EC measurements during the Melpitz campaigns. BS was responsible for





the EC measurements during the Cabauw campaign. REP performed the data analysis, interpreted the results and wrote the manuscript together with RLM and MGB. All co–authors reviewed and commented the manuscript.

**Competing interests**

The authors declare that they have no conflict of interest.



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





Table 1: Overview of reported differences between EC calculated with other protocols minus the EC calculated with the EUSAAR–2 protocol: (a) Piazzalunga et al. (2011); (b) Panteliadis et al. (2015); (c) Giannoni et al. (2016); (d) Han et al. (2016); (e) Cheng et al. (2012); (f) Maenhaut et al. (2012); (g) Karanasious et al. (2015); (h) Cheng et al. (2013); *relative difference compared to EU, a protocol very similar to EUSAAR–2.

| Protocols (TOT) | Relative difference compared to EUSAAR–2 (TOT) |
|---|---|
| NIOSH–870 | ~ –20 % (b) ~ –20,–40 % (~ –10 % washed samples) (c) |
| HE–870 (proxy of NIOSH–870) | ~ –49 % (~ –24 % washed samples) (a) |
| NIOSH–900 | ~ –20, –30, –70, –170 % (f, g) |
| IMPROVE | ~ +25 % (d) |
| IMPROVE–A | ~ –10 % (h) |
| He–580 (proxy of IMPROVE–A) | ~ +6 % (+12 % washed samples) (a); ~ +20 % (e)* |
| EnCan–Total–900 | ~ ±25 % (g) |

Table 2: Description of the methodology for EC mass concentration measurements: thermal protocol, sampling duration, inlet size cut, flow rate and performance during the ERLAP intercomparison, in relative terms for EC bias and variability (Sect. 2.2.3 and Eqs. S1 and S2).

| Station code | Thermal protocol for $m_{EC}$ | Sampling duration (h) | Inlet size cut | Flow rate | EC bias | EC variability |
|---|---|---|---|---|---|---|
| SIR | EUSAAR–2 (TOT) | 12 | $PM_{2.5}$ | 30 m³ in 12 hours | –6.0 % | 11.9 % |
| MEL_W | EUSAAR–2 (TOT) | 24 | $PM_{2.5}$ | 30 m³ h⁻¹ | –6.0 % | 4.3 % |
| MEL_S | EUSAAR–2 (TOT) | 24 | $PM_{2.5}$ | 30 m³ h⁻¹ | 16.1 % | 4.4 % |
| CBW | EUSAAR–2 (TOT) | 12 | $PM_{10}$ | 27.6 m³ in 12 hours | –6.7 % | 3.0 % |
| BOL | EUSAAR–2 (TOT) | 24 | $PM_{2.5}$ | 38.3 L min⁻¹ | 1.6 % | 6.8 % |







Table 3: Description of the measurement methodology for rBC mass concentration adopted during each campaign, along with SP2 owner (Paul Scherrer Institute (PSI), Alfred Wegner Institut (AWI) and Institut de Géosciences de l'Environnement (IGE), calibration material (fullerene soot batch), calibration method selection, and inlet size cut.

| Station code / campaign | SP2 owner | Revision, acquisition card type | Calibration material (fullerene soot batch) | Size selection method for calibration | Inlet size cut | SP2 upper detection limit [nm] |
|---|---|---|---|---|---|---|
| SIR | PSI | C, 14 bits – 2.5 MHz – 8 channels | Fullerene Soot (stock 40971, lot FS12S011) | DMA | PM$_{10}$ | 439 |
| MEL winter | AWI | C, 14 bits – 2.5 MHz – 8 channels | Fullerene Soot (stock 40971, lot W08A039) | APM | PM$_{10}$ | 722 |
| MEL summer | PSI | C, 14 bits – 2.5 MHz – 8 channels | Fullerene Soot (stock 40971, lot FS12S011) | APM | PM$_{10}$ | 766 |
| CBW | IGE | C, 14 bits – 2.5 MHz – 8 channels | Fullerene Soot (stock 40971, lot FS12S011) | DMA | PM$_{10}$ | 537 |
| BOL | PSI | C, 14 bits – 2.5 MHz – 8 channels | Fullerene Soot (stock 40971, lot FS12S011) | DMA | No size cut | 676 |

Table 4: Estimates of potentially missed rBC mass for the two methods ($\Delta m^{\mathrm{extrap}}$ and $\Delta m^{\mathrm{fit}}$), summands contributing to it ($\Delta m_{\mathrm{rBC}>UDL}$ and $\Delta m_{\mathrm{fitresid}}$), and modal diameter of the averaged rBC mass size distribution, all separately listed for each campaign. The size range of 80–300 nm rBC mass equivalent diameter was chosen for fitting the measurement. The sensitivity of the results to this choice was negligible, as discussed in the text.

| Mass fractions/ Campaigns | $\Delta m^{\mathrm{extrap}}$ [%] ( $:= \Delta m_{\mathrm{rBC}<LDL}$) | $\Delta m^{\mathrm{fit}}$[%] | $\Delta m_{\mathrm{rBC}>UDL}$[%] | $\Delta m_{\mathrm{fitresid}}$[%] | $D_{\mathrm{rBC,mode}}$ [nm] |
|---|---|---|---|---|---|
| BOL | 24.1 ± 6.4 | 17.8 ± 7.4 | 0.2 ± 0.3 | −6.5 ± 1.9 | 118.6 ± 0.3 |
| CBW | 22.4 ± 4.9 | 19.7 ± 4.6 | 0.7 ± 0.3 | −3.4 ± 1.0 | 127.2 ± 0.4 |
| MEL summer | 24.5 ± 8.5 | 20.5 ± 7.3 | 0.4 ± 0.3 | −4.5 ± 3.7 | 142.9 ± 0.3 |
| MEL winter | 2.9 ± 2.1 | 2.5 ± 1.4 | 1.1 ± 0.4 | −1.5 ± 1.5 | 227.9 ± 0.7 |
| SIR | 20.3 ± 8.7 | 18.0 ± 7.0 | 2.5 ± 2.0 | −4.9 ± 2.1 | 136.7 ± 0.3 |








Table 5: $m_{rBC}$ and $m_{EC}$ statistics per campaign: median, arithmetic and geometric mean, geometric standard deviation (GSD), standard deviation (SD), $10^{th}$ and $90^{th}$ percentiles and number of data points.

|  | SIR | CBW* | MEL summer | MEL winter | BOL | All the campaigns |
|---|---|---|---|---|---|---|
| $m_{rBC}$ median ($10^{th}$, $90^{th}$) [µg m$^{-3}$] | 0.85 (0.34, 1.55) | 0.47 (0.27, 0.83) | 0.17 (0.09, 0.29) | 1.41 (0.29, 3.56) | 0.44 (0.26, 0.61) | 0.41 (0.13, 1.44) |
| $m_{EC}$* median ($10^{th}$, $90^{th}$), [µg m$^{-3}$] | 0.71 (0.32, 1.36) | 0.92 (0.45, 1.47) | 0.19 (0.11, 0.31) | 0.92 (0.45, 1.47) | 0.56 (0.45, 1.01) | 0.47 (0.14, 1.44) |
| $m_{rBC}$ geometric mean (GSD), [µg m$^{-3}$] | 0.77 (1.88) | 0.46 (1.62) | 0.17 (1.57) | 1.20 (2.58) | 0.40 (1.46) | 0.41 (2.60) |
| $m_{EC}$ geometric mean (GSD), [µg m$^{-3}$] | 0.68 (1.83) | 0.86 (1.63) | 0.18 (1.54) | 0.97 (2.16) | 0.64 (1.45) | 0.47 (2.46) |
| $m_{rBC}/m_{EC}$ geometric mean (GSD) | 1.13 (1.40)** | 0.53 (1.19) | 0.92 (1.26) | 1.23 (1.32) | 0.63 (1.23) | 0.88 (1.50) |
| $m_{rBC}/m_{EC}$ arithmetic mean (SD) | 1.20 (0.51) | 0.54 (0.11) | 0.95 (0.24) | 1.28 (0.33) | 0.65 (0.14) | 0.96 (0.41) |
| $m_{rBC}/m_{EC}$ median ($10^{th}$, $90^{th}$) | 1.20 (0.72, 1.50) | 0.53 (0.44, 0.64) | 0.97 (0.63, 1.23) | 1.29 (0.76, 1.58) | 0.65 (0.49, 0.82) | 0.92 (0.51, 1.42) |
| # data points | 39 | 32 | 55 | 21 | 7 | 154 |

\* EC mass was measured in PM$_{2.5}$, except for the Cabauw campaign, where PM$_{10}$ samples were collected.

\*\* The statistics of the rBC to EC mass ratio for the SIRTA campaign is strongly influenced by one outlier (see Fig. 3). Ignoring this outlier would provide a geometric mean ratio of 1.09 and a GSD of 1.32.



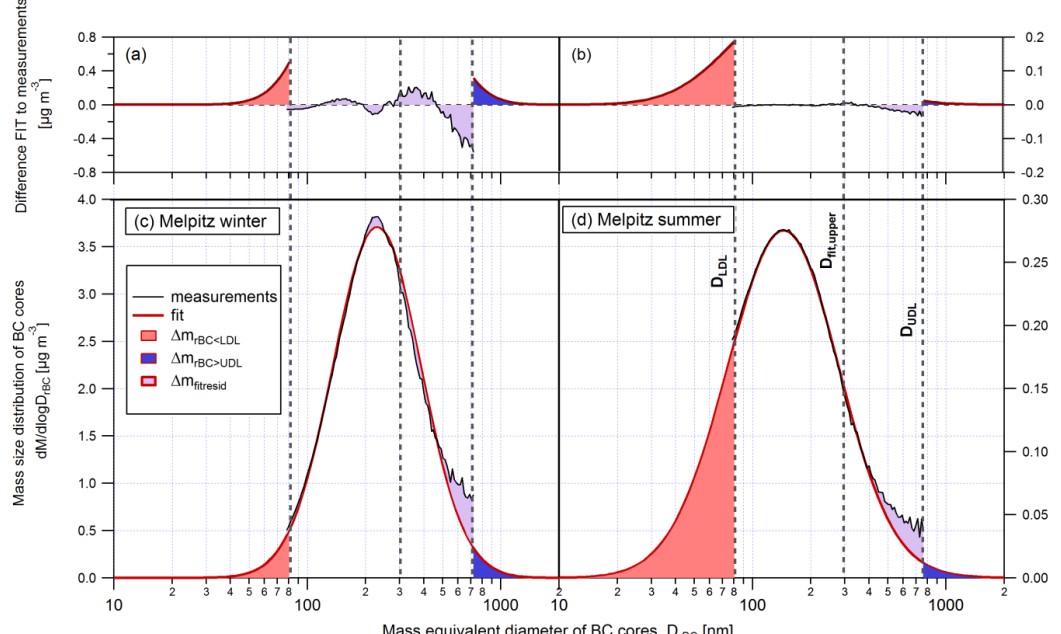

Figure 1: Approach to correct for the rBC mass outside the rBC core size range covered by the SP2 for the Melpitz winter (panels a and c) and the Melpitz summer (panels b and d) campaigns. The bottom two panels show the measured rBC mass size distribution as a function of rBC core mass equivalent diameter, including the SP2 detection limits $D_{LDL}$ and $D_{UDL}$. The lognormal functions are fitted between $D_{LDL}$ and $D_{fit,upper}$. The integrated area of the red, purple, and blue shadings correspond to $\Delta m_{rBC<LDL}$, $\Delta m_{fitresid}$ and $\Delta m_{rBC>UDL}$, respectively (see Sect. 2.3.5). The top two panels additionally show the same shadings after subtraction of the measured size distribution (and measurement forced to be zero outside the SP2 detection range). The average mass size distributions of the other campaigns are represented in Fig. S1.

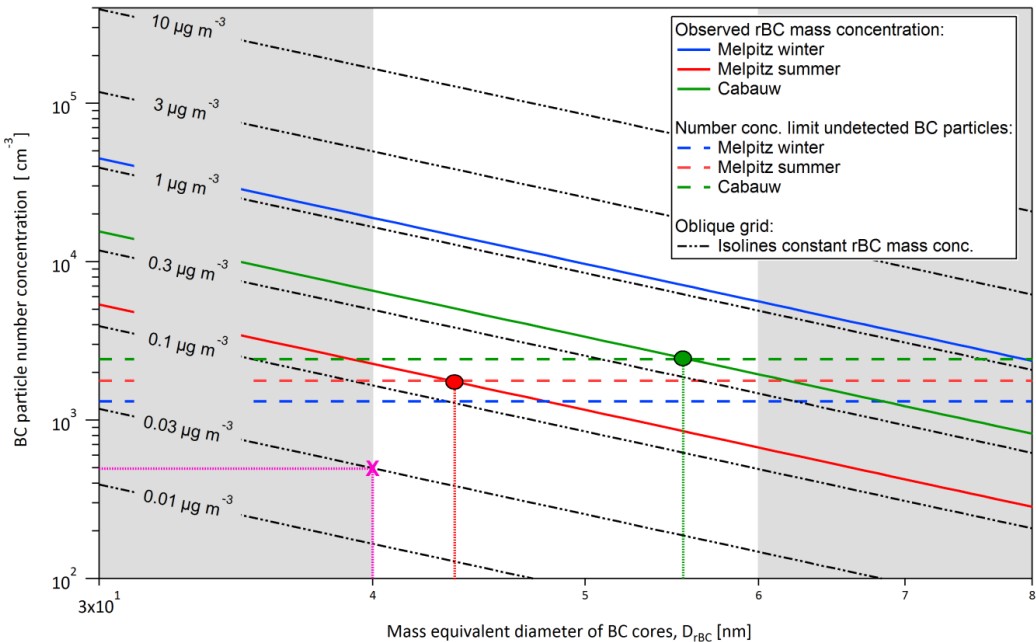

Figure 2: Relationship between rBC mass equivalent diameter, BC particle number concentration and rBC mass concentration





for perfectly monodisperse BC aerosol (magenta cross and lines illustrate an example of this unambiguous relationship). The oblique dashed–dotted black grid represents isolines of constant rBC mass concentration. The continuous oblique lines represent the observed uncorrected rBC mass concentrations (campaign geometric mean values). The horizontal dashed lines represent the upper number concentration limit ($n_{limit}$), calculated as difference between assumed maximum minus measured

BC particle number concentration. The figure can be read in two ways: the intersects of the horizontal lines with rBC mass concentration isolines provide an upper limit for the maximal undetected rBC mass concentration if the undetected mode peaks at the diameter where the intersect occurs. Alternatively, when the horizontal dashed line crosses the corresponding oblique line of equal color (e.g. red point if we consider Melpitz summer), this corresponds to a maximal contribution of rBC mass concentration in small undetected particles equal to the observed value. If we consider Melpitz summer as an example (red

point) this happens for a BC mass mode of 44 nm. The grey shading indicates that the modal diameter of a hypothetical mode undetected by the SP2 is expected to be between 40 nm and 60 nm.

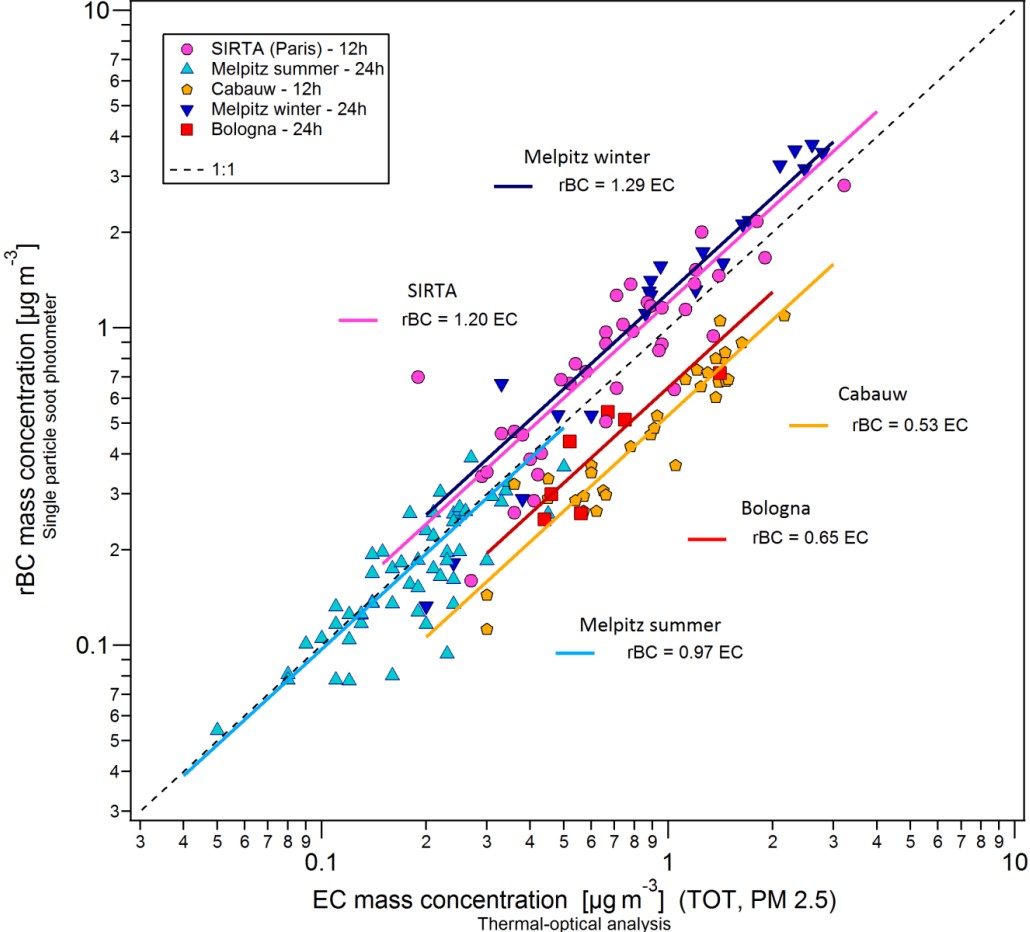


Figure 3: rBC mass concentration versus EC mass concentration for the five campaigns studied in this paper. The median rBC to EC mass ratios are shown as lines for each campaign. Uncertainties of EC measurements as a function of EC and TC filter surface loadings as well as EC/TC mass ratio are presented in Fig. S3 and discussed in Sect. 3.3.2.


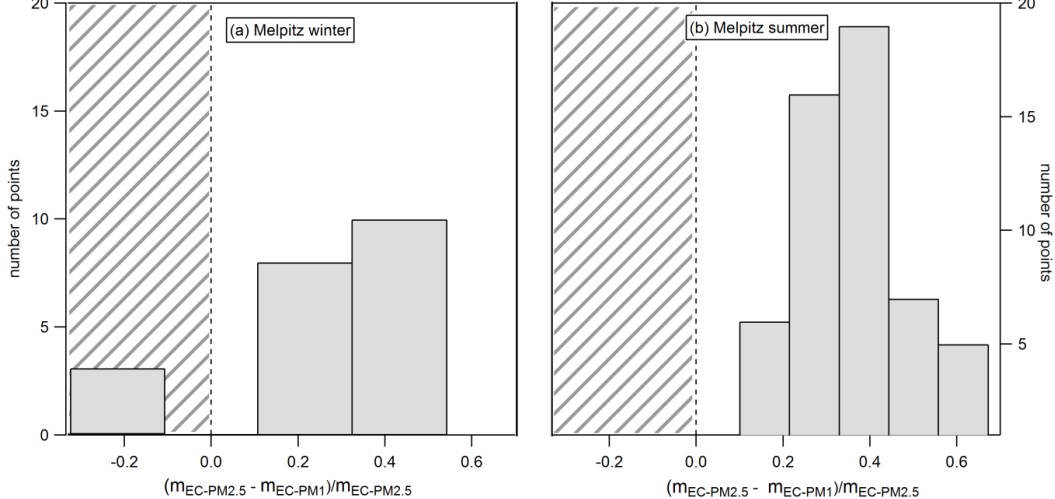

Figure 4: Histogram of $(m_{EC\_PM2.5} - m_{EC_{PM1}})/ m_{EC\_PM2.5}$ for Melpitz winter (panel a, on the left) and summer (panel b, on the right) campaigns. The area with the oblique grey lines indicates the non–physical part in which $m_{EC\_PM2.5} < m_{EC\_PM1}$, reflecting the uncertainty in the EC measurements.


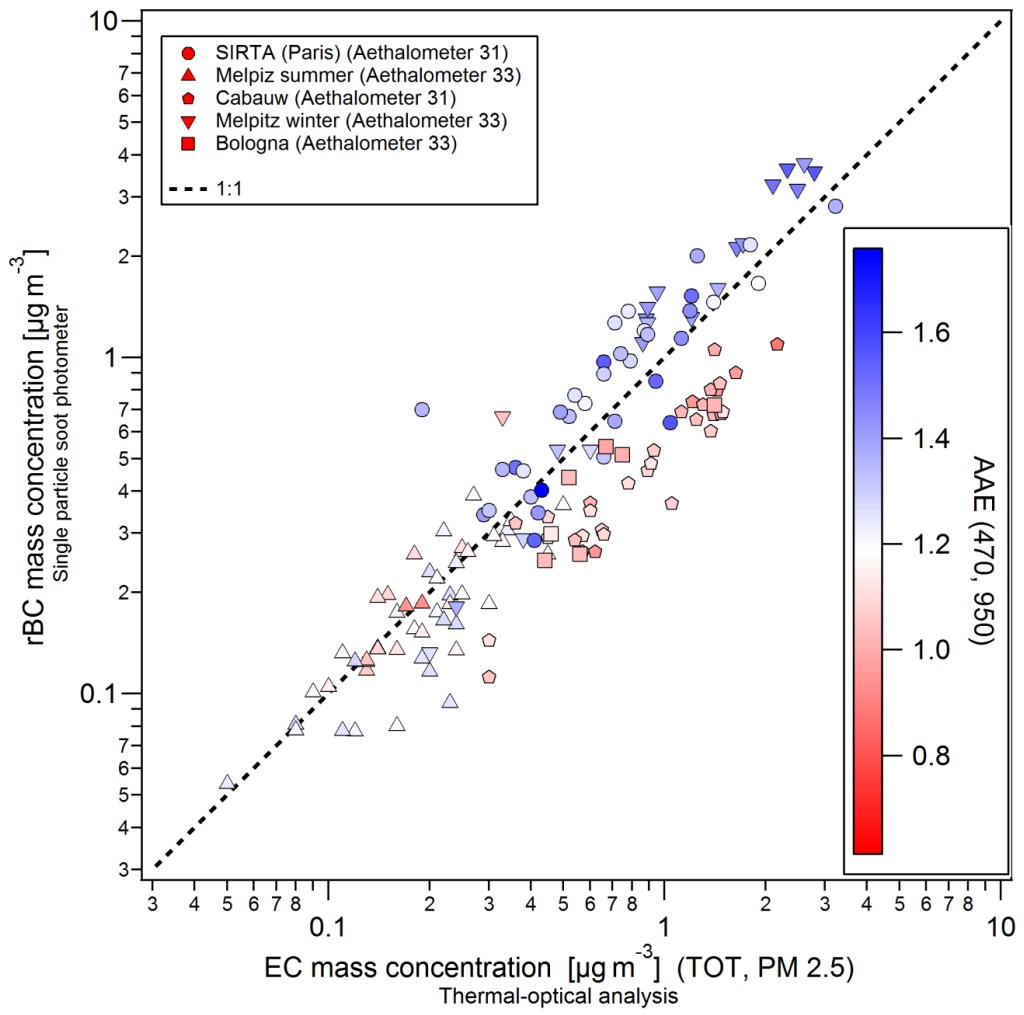

Figure 5: rBC mass concentration versus EC mass concentration for all campaigns of this study color–coded by the Ångström absorption exponent of each data point.




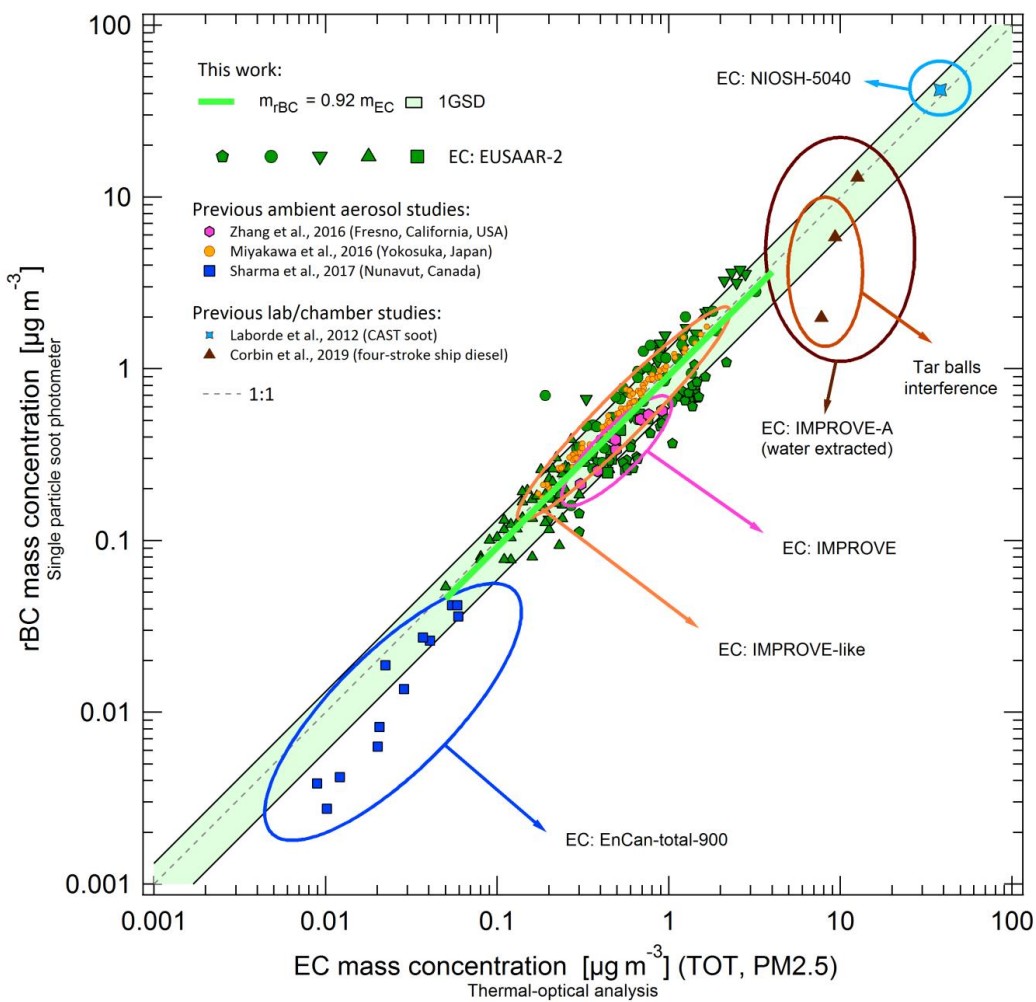

Figure 6: EC mass concentration vs rBC mass concentration for the datasets studied in this work and other published data.

