# Peer review of "Comparison of co–located rBC and EC mass concentration measurements during field campaigns at several European sites"

_Atmospheric Measurement Techniques, 2020_

## Referee Comment (RC1) · Anonymous Referee #1 · 21 Jul 2020

The manuscript reports the results of a general intercomparison of two established yet relatively distinct methods for the analysis (monitoring) of strongly light-absorbing particles (aka soot) in the atmosphere at four different locations in Europe. While the manuscript is based on a large body of measurement data using up-to-date instrumentation and integrating past knowledge into the evaluation of their output, there are a number of concerns that need to be resolved before publication.

The primary objective of the authors is to present a universally valid statistical evaluation of co-located independent measurements of BC (or its surrogates). The issue of light-absorbing carbon in the atmosphere and its measurements is probably one of the

most debated and least resolved problems in atmospheric science. While the authors give a relatively good overview of the specific problems associated with EC/BC measurements, in highlighting the general background (i.e. the nature of light-absorbing carbon in the atmosphere) they fail to account for established prior knowledge in the field. They seem to cherry-pick information from the literature, ignoring important findings that would be needed for better understanding the complexity of the problem and the difficulties with the selected methodologies. For example, the issue of light-absorbing carbon continuum is referred to in the manuscript as being newly discovered (supported by two recent papers of the authors themselves, line 80-85), though it has been on the agenda of aerosol science for nearly 20 years. In general, more focus should be put on this issue, including brown carbon, as this may greatly affect differences between the results of the measurements. Some statements such as 'brown carbon absorbs much less than EC at the red wavelength ($\lambda$ = 635 nm)' (line 211-213) are largely outdated and ignore more recent findings that there are actually two types of brown carbon in aerosol, the strongly absorbing tar balls (carbon spheres) and the weakly absorbing ones (organic chromophores), the former have also been shown to absorb in the near infrared (see e.g Alexander et al. 2008, Saleh et al., 2014, Hoffer et al., 2017).

My another major concern is that even if the authors primarily make 'lump statistics' on all their measurements data, their individual measurement setup is different at each site (e.g. different cutoffs, dryers, SP2 instruments, etc.). In measuring such small-sized and adhesive species even the type and length of the tubing may introduce significant uncertainties due to wall losses, these are likely also different in the measurement sites but not reported here. My fundamental question is the following: if we take into account these uncertainties (or biases?) and the other existing uncertainties (and potential biases) correctly evaluated and reported in the manuscript (I counted 8 significant sources of uncertainties but these may not be all), correctly apply the rules of error propagation, will the overall 8 % difference between the two methods be statistically significant at all? My informed guess is that it will not. So contrary to the key

statement in the manuscript that 'median ratio between observed rBC and EC mass concentrations was 0.92', a more realistic statement would be that 'the two independent methods are indistinguishable within the limits of inherent uncertainties'. Overall, the general impressions from the discussions and conclusions tacitly support this latter statement as the inexplicable variations (to either directions) of campaign-wise data do not reveal any systematic difference between the two methods. This is particularly true since the individual campaigns at the four sites were markedly different in duration (14, 21, 24, 30, 54 days according to my calculations), so the 'mean data' reported in this manuscript refer only to this particular combination. Should the durations of the individual campaigns be different, the overall finding would have been very much different, at least within the limits of extremes. Campaign-wise discussions would have made more sense than calculating a 'European average' value with this fixed setup (which definitely does not exist).

My last major concern is that the authors do not exploit the possibilities of elaborating differences within the individual campaigns (though the campaigns are not too lengthy in themselves). For example, dependence of measured results on trajectory directions at the Melpitz site should have been elaborated to verify the hypothesized effect of coal burning from Eastern Europe. Although these evaluations would not involve statistical analyses, they might still be useful to imply some of the effects that are hypothesized in the manuscript.

---

## Referee Comment (RC2) · Anonymous Referee #2 · 21 Jul 2020

The manuscript fits within the scope of AMT and presents new data on the comparison on EC and rBC measurements. Data are novel but the discussion would need to be improved. The manuscript would be acceptable after major revisions.

Some sections of the manuscript, especially the methods are excessively wordy and could be substantially tightened as too much background information is given that is neither relevant nor appropriate for a method comparison paper. It is unusual in a method section to have paragraphs explaining the basic functioning of commercial instruments (e.g. SP2).

There is also an excessive discussion of artifacts in the thermal methods that does not

really belong here as it does not seem relevant because it is not included in the results discussion. In fact a lot of the discussion is on TOR (reflectance) and the IMPROVE method, when the authors actually use TOT (transmittance). They seem to confuse themselves as in table 1 they refer to IMPROVE protocols as TOT. So this needs to be cleaned up and checked for accuracy. Also ENCan-total-900 (see Sharma et al., ACP, 2017) is neither TOT nor TOR (see your table 1) but I guess you would call it TOA as it has no pyrolysis correction neither by reflectance nor by transmittance. So less text and more accuracy. (BTW quiet a few people also use CTO-375, not mentioned at all)

There is a 7 year delay between the first and the latest studies. With changes in Diesel emission regulations and in car/truck fleet overall in Europe and significant differences between countries France/Germany/Italy... one wonders any impact of this on observations. Still there is no discussion at all on temporal and spatial variability of diesel emissions. Same applies to other "soot" emission sources such as heating, there is only a small discussion on coal. The sites are very different and one would expect different source contributions which will substantially impact results.

Related to the sites. It seems a little "odd" to refer to Paris as a European background site (L22). So I recommend streamlining the names of the sites, so for Paris call it or Paris or Palaiseau but not randomly one or the other plus the site code confuses even more as it is different form the other two. Also overall I am not convinced that it is appropriate to consider Palaiseau as a background site. The same applies to the CNR site in Bologna, which is quite central and not what one would think of as a Po Valley background site. Please be clearer in the description of the sites and the local impact Cabauw is described by its distance to the sea, which is funny when it is closer to both Rotterdam (a major port with related truck traffic) )and Utrecht than the ocean.

A critical scientific issue is artifacts because of particle sizes. This is discussed to a certain extent. However, here again one wonders why there is not more discussion on local sources and differences between sites which will impact particle processing and association of refractory BC and EC with larger (or smaller) particles. That issue is

passed upon. In particular for larger particles and the fact that Cabauw has a PM10 inlet vs a PM2.5.. it misses completely that many studies documented that in processed aerosol EC and BC are associated wo a significant amount with larger particles.

Other (details)

The manuscript preparation could benefit form more attention to detail and is quite careless 2 examples: 1) basic text formatting, starting with the affiliations where none of lines are really aligned in how they start. 2) Melpitz coordinates " MEL; 51° 320' N, 12° 560' E" really?

Please do not use qualitative statements that have no meaning e.g. abstract "the high correlation" what does this mean? Is it statistically significant? Is it not? "high correlation" has no intrinsic meaning. Same in the text.

The abstract is too wordy and especially the second paragraph has no quantitative information is provided. You do not need to have a 3 paragraph abstract.

Your referencing is not very up to date. Many recent papers addressed EC and BC optical properties especially relative to the aethalometer including brown carbon and how this relates to SOA and biomass burning, at the wavelengths used! Please update your referencing and include recent work insights there in your discussion.

You cite so many thermal methods that are not really used. Hardly anybody in air pollution uses the actual NIOSH protocols, nor the Birch and Cary 96, while they are cited, people used variable timesteps or for the least longer time steps. Also the air pollution community hardly ever uses the same final temperature level than NIOSH. SO please clean this up for what is actually being used in the community (e.g. table 1)

The whole discussion on "coarse" BC is misleading (late in the manuscript ∼LK500). In the air quality/aerosol community, coarse tends to mean something very specific: particles between PM2.5 and PM10, sometimes particles larger than PM10 but never to my knowledge particles between PM1 and PM2.5 as it is used here. So or clearly

define but better formulate this differently because it crates confusion given that you have PM10 and PM2.5 size cuts too.

---

## Author Response (AR1)

**Answers to referee 1**

**Comparison of co–located rBC and EC mass concentration measurements during field campaigns at several European sites**

We thank the reviewer for the timely and constructive review of our manuscript, particularly during these difficult corona times. Please find below reviewer comments repeated in black text and our responses in blue text.

Updated responses:
The manuscript has now been revised in accordance with our author responses. We now update the author responses listing all the relevant changes in the revised manuscript. These updates as well as the listed changes are given in red text.

1) While the authors give a relatively good overview of the specific problems associated with EC/BC measurements, in highlighting the general background (i.e. the nature of light-absorbing carbon in the atmosphere) they fail to account for established prior knowledge in the field. They seem to cherry-pick information from the literature, ignoring important findings that would be needed for better understanding the complexity of the problem and the difficulties with the selected methodologies.

The level of methodological detail to be provided in such a manuscript comparing two existing methods remains a subjective choice as the useful level always depends on the reader's prior knowledge on one and/or the other method. Based on the answers given below in response to the referee's specific comments, it appears that the impression of "cherry-picking" was partially caused by keeping artefacts of lesser importance in Sect. 2, while addressing the more relevant ones in Sect. 3. We aim at clarifying these aspects with the revised manuscript.

The specific changes made in response to this general comment are described in the relevant specific comments below.

For example, the issue of light absorbing carbon continuum is referred to in the manuscript as being newly discovered (supported by two recent papers of the authors themselves, line 80-85), though it has been on the agenda of aerosol science for nearly 20 years.

It was by no means our intention to imply "newly discovered" with our statement "refined classification", nor was it sold like that in Corbin et al. (2019). The submitted text referenced by the reviewer is:

> *Recently, Corbin et al. (2019) proposed a refined classification of light–absorbing carbonaceous PM into four classes: soot–BC, char BC, tar brown carbon and soluble brown carbon, and they provided an overview of the respective physico–chemical properties. This refined classification provides a useful framework in describing the responses of TOA and LII. For example tar brown carbon, an amorphous form of carbon, is sufficiently refractory to contribute to EC mass, whereas it is not sufficiently refractory to cause substantial interference in rBC (Corbin and Gysel–Beer, 2019).*

Here, and in our earlier work, we emphasize the concept of a refined classification with respect to measurements. Earlier excellent work is extensively cited in our recent papers. We agree with the reviewer that earlier papers such as that by Bond (2001) have already raised this issue. However, the terms tar and brown carbon were not used by such earlier work, yet have become very popular in recent years (e.g. the next comment by this reviewer). Therefore, in our discussion of brown

carbon, we have cited our recent work rather than the earlier work.

We agree that we missed the opportunity to cite original work such as Bond (2001) at the appropriate place, and we will add a citation to that landmark paper in the revised manuscript.

Bond, T.C. (2001). Spectral dependence of visible light absorption by carbonaceous particles emitted from coal combustion. *Geophys. Res. Lett.,* 28(21):4075–4078.

Update:

The following sentence has been added to the Introduction: "Building on earlier studies (e.g. Bond 2001), Corbin et al. (2019) recently proposed a refined classification of light–absorbing carbonaceous PM into four classes: soot–BC, char BC, tar brown carbon and soluble brown carbon, and they provided an overview of the respective physico–chemical properties"

Bond (2001) is now also cited in Sect. 2.2.1.

In general, more focus should be put on this issue, including brown carbon, as this may greatly affect differences between the results of the measurements. Some statements such as 'brown carbon absorbs much less than EC at the red wavelength ($\lambda = 635$ nm)' (line 211-213) are largely outdated and ignore more recent findings that there are actually two types of brown carbon in aerosol, the strongly absorbing tar balls (carbon spheres) and the weakly absorbing ones (organic chromophores), the former have also been shown to absorb in the near infrared (see e.g Alexander et al. 2008, Saleh et al., 2014, Hoffer et al., 2017).

We agree that tar brown carbon does absorb light also at red and NIR wavelengths. Our statement was not clear enough, and was intended to be 'brown carbon absorbs much less **per unit mass** than EC at the red wavelength of the laser…'. We will add the text in bold.

Update:

The full updated sentence in Sect. 2.2.1 now reads: "However, brown carbon absorbs much less per unit mass than EC at the red wavelength ($\lambda = 635$ nm) of the laser used in the thermal–optical instruments, since its absorbance decreases strongly from the blue–UV region of the electromagnetic spectrum towards the red region (Karanasiou et al., 2015), thereby reducing the potential impact of brown carbon interference"

2) My another major concern is that even if the authors primarily make 'lump statistics' on all their measurements data, their individual measurement setup is different at each site (e.g. different cutoffs, dryers, SP2 instruments, etc.). In measuring such small-sized and adhesive species even the type and length of the tubing may introduce significant uncertainties due to wall losses, these are likely also different in the measurement sites but not reported here.

Diffusion losses are often an important source of error for particle number related quantities, whereas impact on particle mass related quantities is typically smaller. Indeed, diffusion losses were estimated to be less than 10% across the size range of the SP2 (see Sect. 3.3.1), mainly associated with using a dryer in the sampling lines (see Sect. S4).

3) My fundamental question is the following: if we take into account these uncertainties (or biases?) and the other existing uncertainties (and potential biases) correctly evaluated and reported in the manuscript (I counted 8 significant sources of uncertainties but these may not be all), correctly apply the rules of error propagation, will the overall 8 % difference between the two methods be statistically significant at all? My informed guess is that it will not.

So contrary to the key statement in the manuscript that 'median ratio between observed rBC and

EC mass concentrations was 0.92', a more realistic statement would be that 'the two independent methods are indistinguishable within the limits of inherent uncertainties'. Overall, the general impressions from the discussions and conclusions tacitly support this latter statement as the inexplicable variations (to either directions) of campaign-wise data do not reveal any systematic difference between the two methods. This is particularly true since the individual campaigns at the four sites were markedly different in duration (14, 21, 24, 30, 54 days according to my calculations), so the 'mean data' reported in this manuscript refer only to this particular combination. Should the durations of the individual campaigns be different, the overall finding would have been very much different, at least within the limits of extremes. Campaign-wise discussions would have made more sense than calculating a 'European average' value with this fixed setup (which definitely does not exist).

The quote made at the beginning only includes a half sentence taken from the abstract. We always provide the geometric standard deviation along with the geometric mean ratio, also in the abstract, both for the overall and the campaign-wise data sets. For this 0.92 ratio, the GSD covering 68% of the data points was a factor of 1.5. Hence, we believe that the abstract taken as a whole conveys the same message the reviewer is suggesting, and in a quantitative manner.
We also provide recommendations on how to design future experiments towards achieving tighter error margins (e.g. at the end of Sect. 3.3.1 and in the conclusions): "For future intercomparison studies, it is important to constrain the upper cut–off and potential inlet losses of both methods in such a manner that these can be excluded as a source of discrepancy."

4) My last major concern is that the authors do not exploit the possibilities of elaborating differences within the individual campaigns (though the campaigns are not too lengthy in themselves). For example, dependence of measured results on trajectory directions at the Melpitz site should have been elaborated to verify the hypothesized effect of coal burning from Eastern Europe. Although these evaluations would not involve statistical analyses, they might still be useful to imply some of the effects that are hypothesized in the manuscript.

The referee can be assured that we spent plenty of effort trying to understand any potential reason for systematic deviations between the two methods, be it between different campaigns or within a campaign for different air mass origin or variable contributions of sources. This includes:

- Differences in upper size cut-off between the two methods.
- Undetected rBC mass below the lower size cut-off.
- Source specific sensitivity or artefacts of one or both methods.
- Known artefacts or interferences from e.g. tar brown carbon or instrument operation outside parameter range ensuring optimum performance.

Interpretation of these exercises turned out to be a challenging trade-off between providing as much insight as possible while avoiding over-interpretation. As an example: The back trajectory analysis was done for the Melpitz winter example raised by the referee, and we discussed in the last paragraph of Sect. 3.1.1 how coal combustion influence from Eastern Europe likely contributed to a shift to larger BC core sizes and hence reduced missing rBC mass below the lower LOQ of the SP2. At the same time, this shift towards larger BC particle sizes resulted in above-average fraction of total EC in PM2.5 found between 1.0 and 2.5 µm aerodynamic diameter (see Sect. 3.3.1), thereby increasing comparison uncertainties associated with different upper cut-off.

**Answers to referee 2**

1) Some sections of the manuscript, especially the methods are excessively wordy and could be substantially tightened as too much background information is given that is neither relevant nor appropriate for a method comparison paper. It is unusual in a method section to have paragraphs explaining the basic functioning of commercial instruments (e.g. SP2).

The methods section will be shortened.

Update:
The methods section was shortened in the following ways:
- A new supplementary section was created and titled "S1: Further details concerning the optical correction in thermal optical analysis (TOA)". Some information that was previously in Sect. 2.2.1 have been moved into this Sect. S1. The main text refers to this new Section in the following way "The measurement principle behind this so-called thermal-optical transmission (TOT) correction approach is explained in Sect. S1. The charring correction can also be done using light reflectance (thermo–optical reflectance method; TOR) instead of transmittance. As reported in the review paper by Karanasiou et al. (2015), EC values of atmospheric samples determined using the TOT method are often up to 30–70 % lower than those determined using the TOR method due to greater evaporation and saturation artefacts in the reflection approach (see Sect. S1). Therefore, all EC mass values reported in this study are based on the TOT method.".
- Following the above change, a paragraph was moved from Sect. 2.2.1 to 2.2.2 in order to streamline the discussion. Sect. 2.2.2 is now titled "Thermal protocols: EUSAAR–2 vs other existing protocols".
- A paragraph containing basic details about the SP2 in Sect. 2.3.1 was deleted.

2) There is also an excessive discussion of artifacts in the thermal methods that does not really belong here as it does not seem relevant because it is not included in the results discussion. In fact a lot of the discussion is on TOR (reflectance) and the IMPROVE method, when the authors actually use TOT (transmittance). They seem to confuse themselves as in table 1 they refer to IMPROVE protocols as TOT. So this needs to be cleaned up and checked for accuracy. Also ENCan-total-900 (see Sharma et al., ACP, 2017) is neither TOT nor TOR (see your table 1) but I guess you would call it TOA as it has no pyrolysis correction neither by reflectance nor by transmittance. So less text and more accuracy. (BTW quite a few people also use CTO-375, not mentioned at all).

The discussion will be shortened retaining focus on the methods applied in this or previous rBC/EC intercomparison studies. TOT/TOR information provided in Table 1 will be corrected. Table 1 addresses all protocols applied in studies included in Fig. 6, and CTO-375 protocol was not mentioned, as it is not among these. Anyway, Table 1 and Section 2.2.2 will be shortened in response to comment 10) given below.

Update:
As explained in the preceding response to comment #1 many of the details regarding the optical correction have now been moved to Supplementary Text S1.
Table 1 has been shortened considerably and the TOT/TOR information corrected. The revised Table 1 is copied here:

Table 1: Overview of reported differences between EC calculated with other protocols minus the EC calculated with the EUSAAR–2 protocol: (a) Han et al. (2016); (b) Cheng et al. (2013); (c) Karanasiou et al. (2015).

| Protocols | Relative difference compared to EUSAAR–2 (TOT) |
|---|---|
| IMPROVE (TOR) | ~ +25 % (a) |
| IMPROVE–A (TOR) | ~ –10 % (b) |
| EnCan–Total–900 | ~ ±25 % (c) |

3) There is a 7 year delay between the first and the latest studies. With changes in Diesel emission regulations and in car/truck fleet overall in Europe and significant differences between countries France/Germany/Italy... one wonders any impact of this on observations. Still there is no discussion at all on temporal and spatial variability of diesel emissions. Same applies to other "soot" emission sources such as heating, there is only a small discussion on coal. The sites are very different and one would expect different source contributions, which will substantially impact results.

The molecular structure of BC emitted from diesel engines can vary between different engines and engine operation conditions. This variation could potentially affect either method as reactivity and optical properties of BC are related to it. In particular, BC with a lower degree of graphitization is more easily oxidized in the diesel particle filters (DPF) included in modern diesel engines (e.g Schmid et al., 2011). However, emissions from more modern cars equipped with DPF are marginal compared to older cars without DPF, resulting in a decreasing trend of traffic BC emissions and a corresponding shift towards larger relative contributions from other BC sources as a potential longer-term trend in the observed ratios. Furthermore, seasonality and regional differences are at least as important, as seen from clear differences between the BC particle properties and intercomparison results for the Melpitz summer and winter campaigns. We used back trajectory analysis and the "aethalometer model" to assess potential systematic relations between sources and observed ratios (Sect. 3.3.3); however, the results were not conclusive for the reason stated in the conclusions: "The discrepancy between rBC and EC appears to be systematically related to the BC source, i.e. traffic versus wood and/or coal burning. However, it was not possible to identify causalities behind this trend due to potential cross–correlations between several aerosol and BC properties relevant for potential biases." - It is unrealistic to expect that the available data set would provide the basis to demonstrate a fleet modernization effect and hence we refrain from undertaking such an attempt.

Schmid, J., Grob, B., Niessner, R., and Ivleva, N. P.: Multiwavelength Raman microspectroscopy for rapid prediction of soot oxidation reactivity, Anal. Chem., 83, 1173–1179, doi:10.1021/ac102939w, 2011.

4) Related to the sites. It seems a little "odd" to refer to Paris as a European background site (L22). So I recommend streamlining the names of the sites, so for Paris call it or Paris or Palaiseau but not randomly one or the other plus the site code confuses even more as it is different form the other two. Also overall I am not convinced that it is appropriate to consider Palaiseau as a background site. The same applies to the CNR site in Bologna, which is quite central and not what one would think of as a Po Valley background site. Please be clearer in the description of the sites and the local impact Cabauw is described by its distance to the sea, which is funny when it is closer to both Rotterdam (a major port with related truck traffic) and Utrecht than the ocean.

The wording chosen in the abstract was indeed imprecise, whereas the classification provided along with the site descriptions in Sect. 2.1 is appropriate (i.e. "suburban background" and "urban background" for SIRTA and CNR Bologna, respectively). We will replace "SIRTA" by Palaiseau throughout the manuscript. As for Cabauw, we will add the distances to the Rotterdam and Utrecht to the site description.

Update:
"SITRA" was replaced by "Palaiseau" at all the relevant places in the manuscript, including in the text, in the tables, and in many of the figures.
Regarding Cabauw, its site description has been updated to the following: "The KNMI (Koninklijk Nederlands Meteorologisch Instituut) Cabauw Experimental Site for Atmospheric Research (Netherlands; 51° 58' N, 4° 55' E, –0.7 m a.s.l.) is located in the background area of Cabauw, 20 km from Utrecht, 30 km from Rotterdam and 50 km from the North sea."

**5)** A critical scientific issue is artifacts because of particle sizes. This is discussed to a certain extent. However, here again one wonders why there is not more discussion on local sources and differences between sites which will impact particle processing and association of refractory BC and EC with larger (or smaller) particles. That issue is passed upon. In particular for larger particles and the fact that Cabauw has a PM10 inlet vs a PM2.5. it misses completely that many studies documented that in processed aerosol EC and BC are associated to a significant amount with larger particles.

Several processes alter the size distribution of BC containing particles. For example condensation of secondary particulate matter or coagulation with BC-free particles will increase the aerodynamic diameter, whereas the BC mass equivalent diameter, which determines the SP2 upper cut-off, remains unaffected by condensation. Some studies indicate a shift towards smaller BC core diameters during transport with precipitation due to preferential wet-removal of larger particles. As for our study, we have provided a quite extensive discussion on upper cut-off effects in Sect. 3.3.1. Cabauw was among the sites with a small modal size in terms of rBC core mass equivalent diameter, which does not exclude a second mode of supermicron BC cores. For Melpitz, where EC measurements were available with different PM cut-off diameters, we tried to identify a relationship between supermicron EC fraction and rBC vs EC discrepancy - without a significant result. It is correct that this question is passed upon as we clearly state in the conclusions: "The discrepancy between rBC and EC appears to be systematically related to the BC source, i.e. traffic versus wood and/or coal burning. However, it was not possible to identify causalities behind this trend due to potential cross–correlations between several aerosol and BC properties relevant for potential biases. For future intercomparison studies, it is important to constrain the upper cut–off and potential inlet losses of both methods in such a manner that these can be excluded as a source of discrepancy."

**Other (details)**

**6)** The manuscript preparation could benefit form more attention to detail and is quite careless 2 examples: 1) basic text formatting, starting with the affiliations where none of lines are really aligned in how they start. 2) Melpitz coordinates "MEL; 51∘ 320' N, 12∘ 560' E" really?

These technical edits will be implemented and typos corrected.

Update:
We have carefully checked the manuscript formatting. The author affiliations and site coordinates have all been updated as suggested.

**7)** Please do not use qualitative statements that have no meaning e.g. abstract "the high correlation" what does this mean? Is it statistically significant? Is it not? "high correlation" has no intrinsic meaning. Same in the text.

We will replace "high correlation" with more precise statements along the line: "…the observed correlation between rBC and EC mass reveals a linear relationship with a constant ratio, thus providing clear evidence that both methods essentially quantify the same property of atmospheric aerosols…"

Update:
The relevant sentence in the abstract now reads: "Overall, the observed correlation between rBC and EC mass reveals a linear relationship with a constant ratio, thus providing clear evidence that both methods essentially quantify the same property of atmospheric aerosols, whereas systematic differences in measured absolute values by up to a factor of 2 can occur."

**8)** The abstract is too wordy and especially the second paragraph has no quantitative information is provided. You do not need to have a 3 paragraph abstract.

The abstract will be shortened.

Update:
The abstract has been shortened. The 2nd paragraph in the abstract has been deleted, with only a minor part being moved to the 1st paragraph.

9) Your referencing is not very up to date. Many recent papers addressed EC and BC optical properties especially relative to the aethalometer including brown carbon and how this relates to SOA and biomass burning, at the wavelengths used! Please update your referencing and include recent work insights there in your discussion.

Our manuscript does not deal with comparing optically derived equivalent black carbon, eBC, with EC and/or rBC. Spectral dependence of brown carbon optical properties come into play for the optical charring correction. This is, however, only of secondary importance for the intercomparison results of this study. More relevant is the refractoriness and reactivity, which is the primary criterion affecting the split between BC and other particulate matter in either method. "Brownness" and refractoriness are related to each other for most carbonaceous materials, whereas the latter is of greater importance in the context of this study. We will modify the text, also in response to comments by the first referee, to further clarify these aspects including additional references.

Update:
The relevant changes to the manuscript are described above in response to comment number 1 from Reviewer 1.

10) You cite so many thermal methods that are not really used. Hardly anybody in air pollution uses the actual NIOSH protocols, nor the Birch and Cary 96, while they are cited, people used variable timesteps or for the least longer time steps. Also the air pollution community hardly ever uses the same final temperature level than NIOSH. SO please clean this up for what is actually being used in the community (e.g. table 1)

We cited Birch and Cary 96 in the context of the introduction and basics of the NIOSH technique, but we acknowledge the reviewer's points here. The discussion of NIOSH protocols was motivated by the fact that Fig. 6 includes a NIOSH-5040 based data point. However, in the interest of shortening the methods part, as requested above, we will shorten Table 1 and Section 2.2.2.

Update:
The changes made to Table 1 and Section 2.2 are described above in response to comments number 1 and 2 from Reviewer 2.

11) The whole discussion on "coarse" BC is misleading (late in the manuscript ~LK500). In the air quality/aerosol community, coarse tends to mean something very specific: particles between PM2.5 and PM10, sometimes particles larger than PM10 but never to my knowledge particles between PM1 and PM2.5 as it is used here. So or clearly define but better formulate this differently because it crates confusion given that you have PM10 and PM2.5 size cuts too.

The use of "coarse" and "fine" BC in Sect. 3.3.1 will be checked carefully in order to minimize potential confusion.

Update:
In Sect. 3.3.1 coarse particles are now defined specifically as referring to the size fraction between 1 and 2.5 µm, and EC concentrations within this size range are referred to as $EC_{2.5-1}$.

[revised manuscript text omitted]

**Supplementary information (SI)**

**S1. Further details concerning the optical correction in thermal optical analysis (TOA)**

During TOA analysis a fraction of the OC can pyrolyze in the He step to form pyrolytic carbon (PC), which is thermally stable and only desorbs in the $O_2$ step, thereby causing a charring artefact in the mutual quantification of OC and EC. To correct for this latter effect a laser at 658 nm can be used to monitor the light transmission through the loaded filter before and during the analysis. PC is strongly light absorbing, thus leading to a decrease of the transmission signal when it forms upon heating in the inert atmosphere. Later, in the oxidizing atmosphere, both PC and EC are released from the filter resulting in an increase of the transmission signal. The time at which the transmission equals again the initial pre–pyrolysis value is used to separate OC and EC, depending on whether the carbon evolved before or after this "split point", respectively. This thermal–optical transmittance (TOT) approach to correct for PC eliminates potential charring artefacts if the PC has the same mass–specific attenuation cross section as the atmospheric native EC (Yang and Yu, 2002), and if no other light–absorbing material evolves from the sample.

Instead of using light transmission, the charring correction can also be done with light reflectance (i.e., thermal-optical reflectance, TOR). EC values determined using TOT can be up to 30 – 70 % lower than those determined with TOR (Karanasiou et al., 2015), because the evaporation of non–absorbing particulate matter during heating affects the reflectance to a greater extent than the transmission signal. Furthermore, high loadings of EC result in saturation effects of both optical signals, again to a greater extent for the reflection compared to the transmission method (Chiappini et al., 2014). These two effects result in better reproducibility and accuracy of the TOT based OC/EC split compared to the TOR approach.

**S2. Calculation of variability and bias**

The variability ($Q_{AV}$) is defined as the relative standard deviation given by the 95 % confidence limit, thus:

$$Q_{AV} = \frac{n}{\sqrt{6}} \left[ max \left( \frac{RD_i}{T_i} \right) - min \left( \frac{RD_i}{T_i} \right) \right] \qquad (S1)$$

Where $RD_i = L_i - T_i$, with $L_i$ and $T_i$ representing the laboratory and expected concentrations, respectively.

The bias ($Q_{AB}$) is defined as the median of the percentage of the ratio between $RD_i$ and $T_i$

$$Q_{AB} = median \left| \frac{RD_i}{T_i} \% \right|. \qquad (S2)$$

**S3. The Cunningham slip correction**

The Cunningham slip correction factor, $C_C$, is used to account for non–continuum effects when calculating the drag force on small particles. $C_C$ depends on the particle diameter, $D$, the mean free path of the surrounding gas, $\lambda$, and on the experimental coefficients $\alpha$, $\beta$ and $\gamma$ (Cunningham, 1910; Seinfeld and Pandis, 2006).

$$C_C(D) = 1 + \frac{2\lambda}{D} \left[ \alpha + \beta e^{-\frac{\gamma D}{\lambda}} \right] \qquad (S3)$$

with $\alpha = 1.257$, $\beta = 0.4$, $\gamma = 1.1$, $\lambda = 6.5 \ 10^{-8}$ m

**S4. Hygroscopic growth factor**

Hygroscopic growth affects the cut–off imposed by impactors operated at ambient RH. Here we provide simplified equations to calculate the volume equivalent diameter growth factor, GF, of BC–containing particles coated with a mixture of organic and inorganic matter. The GF is calculated using $\kappa$–Köhler theory (Petters and Kreidenweis, 2007):

$$\text{GF(RH)} = (1 + \kappa_{tot} \frac{\text{RH}}{1-\text{RH}})^{1/3} \tag{S4}$$

where the hygroscopicity parameter of the mixed particle, $\kappa_{tot}$, is obtained with the ZSR–mixing rule written as (Petters and Kreidenweis, 2007):

$$\kappa_{tot} = \varepsilon_{BC}\kappa_{BC} + \varepsilon_{org}\kappa_{org} + \varepsilon_{inorg}\kappa_{inorg} \tag{S5}$$

The hygroscopicity parameters $\kappa$ of BC, organics and inorganics are assumed to be 0, 0.1 and 0.5, respectively (Engelhart et al., 2012). The volume fraction, $\varepsilon_x$, of compound class "x" in the particle can be calculated using:

$$\varepsilon_x = \frac{m_x}{m_{tot}} \frac{\rho_{tot}}{\rho_x} \tag{S6}$$

where $m_x$ is the mass of "x" in the particle, $m_{tot}$ is the total particle mass. $\rho_x$ is the material density of "x", which is assumed to be 1800 kg m$^{-3}$, 1200 kg $m^{-3}$ and 1700 kg $m^{-3}$ for BC, organics and inorganics, respectively. The mixed particle density, $\rho_{tot}$, is obtained with:

$$\rho_{tot} = \frac{1}{\frac{m_{BC}}{m_{tot}}\rho_{BC} + \frac{m_{org}}{m_{tot}}\rho_{org} + \frac{m_{inorg}}{m_{tot}}\rho_{inorg}} \tag{S7}$$

**S5. Dryers and losses**

Differences in $m_{rBC}$ and $m_{EC}$ mass concentration can also come from differences in the losses of the respective sampling inlets. Particle losses can be caused by the presence of a dryer in the inlet line. The dryer technology offers different dryer types including diffusion and membrane dryers. Diffusion dryers use chemical adsorbents such as silica gel for the minimization of aerosol losses. When the aerosol passes through the tube, the silica adsorbs water vapor, therefore this chemical needs to be changed and regenerated on a regular basis. Membrane dryers are elastic tubes based on water vapor–permeable polytetrafluoroethylene (PTFE). Commercially available membranes are products such as Nafion®: a sulfonated tetrafluoroethylene working as permeable membrane in which water vapor molecules are transported.

In any case, particle losses by diffusion across the dryer should be accounted for in the data processing. This is done by calculating an equivalent tube length, which is longer than the actual dryer length (Wiedensohler et al., 2012).

In this work we calculate the diffusion losses when dryers were present in the inlet lines. A diffusion dryer was placed in front of the SP2 line during the Bologna campaign; while the EC line had none. The particle transmission efficiency of the diffusion dryer for a flow rate of 2 L min$^{-1}$ was evaluated to be less than 10 % on average on the size range of the SP2 measurements. Nafion dryers (model MD–700, Perma Pure) were set in front of the SP2s in the Melpitz and Cabauw campaigns. The diffusion losses through them were calculated with the hypothesis of laminar flow and were found to be negligible. The particle loss due to the presence of a dryer in the inlet line does

1405  not seem to be the main reason of the $m_{\mathrm{rBC}}$ and $m_{\mathrm{EC\_PM2.5}}$ discrepancies for the campaigns of this study. For this reason we did not correct the rBC mass concentrations for this effect.

**Tables SI**

Table S1: site, station code, coordinates, altitude and year/season of the field campaigns presented in this work.

| Site (country) | Station code | Coordinates | Altitude | Year/season |
|---|---|---|---|---|
| Palaiseau (FR) | Palaiseau | 48.713° N 2.208° E | 160 | 2010/winter |
| Melpitz (DE) | Melpitz | 51°32' N, 12°56' E | 86 | 2017/winter and 2015/summer |
| Cabauw (NE) | Cabauw | 51° 58' N, 4° 55' E | –0.7 | 2016/autumn |
| Bologna (IT) | Bologna | 44° 31' N, 11° 20' E | 39 | 2017/summer |

1410

Table S2: The first three columns show median, 10th and 90th percentiles of EC, TC and EC/TC filter loading. The last four columns show minima and maxima of EC and TC filter loadings for the field campaigns of this study.

| Station code | EC filter loading median (10th, 90th) [$\mu g\ cm^{-2}$] | TC filter loading median (10th, 90th) [$\mu g\ cm^{-2}$] | EC/TC filter loading median (10th, 90th) [$\mu g\ cm^{-2}$] | Min EC filter load [$\mu g\ cm^{-2}$] | Max EC filter load [$\mu g\ cm^{-2}$] | Min TC filter load [$\mu g\ cm^{-2}$] | Max TC filter load [$\mu g\ cm^{-2}$] |
|---|---|---|---|---|---|---|---|
| Palaiseau | 1.66 (0.76, 3.19) | 9.79 (5.02, 24.03) | 0.14 (0.09, 0.30) | 0.44 | 7.53 | 3.72 | 37.56 |
| Melpitz winter | 4.30 (1.50, 11.08) | 44.73 (9.77, 110.72) | 0.11 (0.09, 0.17) | 0.93 | 12.53 | 5.45 | 115.03 |
| Melpitz summer | 0.90 (0.48, 1.43) | 12.47 (6.93, 19.08) | 0.08 (0.05, 0.10) | 0.25 | 2.30 | 4.76 | 24.43 |
| Cabauw | 1.47 (0.71, 2.34) | 6.87 (3.78, 14.95) | 0.18 (0.12, 0.29) | 0.48 | 3.44 | 2.92 | 19.10 |
| Bologna | 2.49 (1.84, 2.66) | 15.50 (13.20, 17.72) | 0.16 (0.11, 0.19) | 1.63 | 2.74 | 12.46 | 18.42 |

Table S3: In this table the aerodynamic particle diameter $D_{aero}$ corresponding to the upper SP2 cut–off, which depends on the BC mass equivalent diameter, $D_{ve}$ (calculating starting from the BC core mass equivalent diameter, $D_{rBC}$, knowing the particle mixing state), is given for different extreme hypotheses concerning particle shape ($\chi$), mixing state and relative humidity (RH). The calculations are made with the hypothesis of fixed density of BC core $\rho_{rBC} = 1800\ kg\ m^{-3}$ and with the hypothesis of coating made by half organic material with $\kappa_{org} = 0.1$ and $\rho_{org} = 1200\ kg\ m^{-3}$ and half inorganic material with $\kappa_{inorg} = 0.5$ and $\rho_{inorg} = 1700\ kg\ m^{-3}$. From these hypothesis and knowing the ratio between the mass of the coating material, $m_{coat}$ and the mass of the BC core, $m_{rBC}$, the total

particle density, $\rho_P$, can be calculated. The mixing state of the last example particle is constrained with SP2 measurements during the Melpitz winter campaign.

| | $D_{rBC}$ [nm] | $\rho_P$ [kg m$^{-3}$] | $\kappa_{tot}$ | $D_{ve}$ [nm] Dry | GF (RH = 80 %) | $D_{ve}$ [nm] (RH = 80 %) | GF (RH = 9 5%) | $D_{ve}$ [nm] (RH = 95 %) | $D_{aero}$ [nm] Dry | $D_{aero}$ [nm] (RH = 80 %) | $D_{aero}$ [nm] (RH = 95 %) |
|---|---|---|---|---|---|---|---|---|---|---|---|
| Fractal–like pure BC: $\chi$ = 2.4 (Park et al., 2003) | 722.0 | 1800.0 | 0.0 | 722.0 | 1.0 | 722.0 | 1.0 | 722.0 | 625.3 | 625.3 | 625.3 |
| Spherical pure BC: $\chi$ = 1 | 722.0 | 1800.0 | 0.0 | 722.0 | 1.0 | 722.0 | 1.0 | 722.0 | 968.7 | 968.7 | 968.7 |
| Coated BC: $\chi$ = 1, $m_{coat}$= $m_{rBC}$ | 722.0 | 1579.4 | 0.1 | 909.7 | 1.2 | 1063.1 | 1.6 | 1423.4 | 1143.2 | 1336.0 | 1788.8 |
| Coated BC: $\chi$ = 1, $m_{coat}$= $6m_{rBC}$ | 722.0 | 1452.2 | 0.2 | 1381.1 | 1.2 | 1722.4 | 1.8 | 2432.5 | 1664.4 | 2075.6 | 2931.3 |
| Coated BC with coating as observed during the Melpitz Winter campaign: $\chi$ = 1, $m_{coat}$= 2.33 [0.99–3.17] $m_{rBC}$ | 722.0 | 1505.2 | 0.2 | 1078.2 | 1.2 | 1310.4 | 1.7 | 1815.9 | 1322.8 | 1607.7 | 2227.9 |

1425    Table S4: AAE (470,950) statistics for the campaigns in this study: median, geometric mean, 10$^{th}$ and 90$^{th}$ percentiles and number of data points.

| | PalaiseauSIR | CabauwCBW | MelpitzMEL summer | MelpitzMEL winter | BolognaBOL |
|---|---|---|---|---|---|
| AAE median (10, 90) | 1.35 (1.24, 1.53) | 1.05 (0.97, 1.12) | 1.19 (1.09, 1.26) | 1.40 (1.28, 1.50) | 1.03 (1.01, 1.07) |
| AAE geometric mean | 1.36 | 1. 04 | 1.18 | 1.38 | 1.04 |
| # points | 34 | 32 | 49 | 20 | 7 |

Table S5: Summary of site name, country, SP2 calibration material, $m_{EC}$ cut–off and TOA thermal protocol, sampling period, site characteristics and geometric mean of the $m_{rBC}/m_{EC}$ ratio for all the data. *One data point

| | SP2 Calibration Material | $m_{EC}$ cut–off/ TOA Thermal technique | Site name, country | Season/year and site characteristics | $m_{rBC}/m_{EC}$ median |
|---|---|---|---|---|---|
| PalaiseauSIR | Fullerene Soot | PM$_{2.5}$/ EUSAAR–2 | PalaiseauParis, France | Jan/Feb 2010 suburban background | 1.20 |
| CabauwCBW | Fullerene Soot | PM$_{10}$/ EUSAAR–2 | Cabauw, Netherlands | Oct 2016 rural background | 0.53 |
| BolognaBOL | Fullerene Soot | PM$_{2.5}$/ EUSAAR–2 | Bologna, Italy | July 2017 urban background | 0.65 |
| MelpitzMEL winter | Fullerene Soot | PM$_{2.5}$/ EUSAAR–2 | Melpitz, Germany | Feb 2017 rural background | 1.29 |
| MelpitzMEL summer | Fullerene Soot | PM$_{2.5}$/ EUSAAR–2 | Melpitz, Germany | July 2015 rural background | 0.97 |
| Zhang et al. (2016) | Fullerene Soot | PM$_{2.5}$/ IMPROVE | Fresno, California, USA | Jan/Feb 2013 urban background | 0.70 |
| Miyakawa et al. (2016) | Fullerene Soot | PM$_{2.5}$/ IMPROVE–like | Yokosuka, Japan | Summer 2014 June 17 – 27 urban | 1.07 |

| Sharma et al. (2017) | Aquadag scaled to Fullerene Soot | PM₁/ EnCan–Total– 900 | Nunavut, Canada | From Mar 2011 to Dec 2013 remote site | 0.55 |
| Corbin et al. (2019) | Fullerene Soot | PM₁/ IMPROVE–A (washed) | – | Chamber study – four–stroke ship diesel engine | 1.03* |
| Laborde et al. (2012b) | Fullerene Soot | NIOSH–5040 | – | Chamber study – CAST soot | 1.10* |

1430

1435

1440 **Figures SI**

[Figure]

[Figure]

Figure S1: Approach to correct for the rBC mass outside the rBC core size range covered by the SP2 for the Bologna (panels a and d), the Palaiseau (panels b and e) and the Cabauw (panels c and f) campaigns. The bottom three panels show the measured rBC mass size distribution as a function of rBC core mass equivalent diameter, including the SP2 detection limits $D_{LDL}$ and $D_{UDL}$. The lognormal functions are fitted between $D_{LDL}$ and $D_{fit,upper}$. The integrated area of the red, purple, and blue shadings correspond to $\Delta m_{rB,<LDL}$, $\Delta m_{fitresid}$ and $\Delta m_{rBC>UDL}$, respectively (see Sect. 2.3.5). The top three panels additionally show the same shadings after subtraction of the measured size distribution (and measurement forced to be zero outside the SP2 detection range).

[Figure]

[Figure]

Figure S2: Statistics (10th, 25th, 50th, 75th and 90th percentiles, arithmetic and geometric means, SD and GSD) of the rBC to EC mass ratio ($m_{rBC}/m_{EC}$) per campaign (panel a) and with all the campaigns of this work (panel b).

1455

[Figure]

1460 Figure S3: rBC mass concentration versus EC filter loading (panel a), TC filter loading (panel b) and EC/TC mass ratio (panel c). The red shaded areas in panel (a) and (b) indicate the high EC surface loading and the low TC surface loading areas respectively.

[Figure]

[Figure]

Figure S4: Relative difference between $m_{rBC}$ and $m_{EC}$ versus the AAE(470,950) coloured by campaign.

1465

---

## Author Response (AR2)

**Answers to referee 1**

**Comparison of co–located rBC and EC mass concentration measurements during field campaigns at several European sites**

We thank the referee for the timely review of our revised manuscript, which helped in further clarifying potential interferences from different types of brown carbon in EC and rBC mass measurements. Please find below reviewer comments repeated in black text, our responses in blue text, and passages in the revised manuscript in red text.

1) Most of the concerns raised in the first review have now been rectified and the manuscript has improved significantly. However, there remain two issues that are not addressed properly. In the Abstract and Conclusion there is the explicit statement that "Overall, considering the five field campaigns, the median of the observed rBC to EC mass ratios for the whole dataset was 0.92, with a GSD of 1.50." While it is true that mathematically the very high GSD practically means that the two methods are indistinguishable within the limits of inherent uncertainties, given the fact that the mean only refer to the specific combination of individual campaigns of markedly different durations (14, 21, 24, 30, 54 days) without a-priori methodological planning and standardization, to provide a mean value is absolutely meaningless. If, for example, the campaign that provides mean value greater than unity had lasted twice as long whereas the other having mean value less than unity had been significantly reduced in duration, a 'mean value' exceeding unity would have been easily obtained (of course, with a similarly large GSD which conveys basically the same mathematical meaning). So I strongly suggest the authors should refrain from reporting 'mean' numbers in the Abstract and Conclusion, because it explicitly implies that the experimental setup had been well designed, standardized and balanced and are able to provide robust and high-quality data for detailed statistical analyses. This was certainly not the case with the manuscript.

We have removed any reference "mean" values from abstract and conclusions.

Abstract:
[…]The observed values of median rBC to EC mass concentration ratios on single campaign level were 0.53, 0.65, 0.97, 1.20 and 1.29, respectively, and the geometric standard deviation (GSD) was 1.5 when considering all data points from all five campaigns. This shows that substantial systematic bias between these two quantities occurred during some campaigns, which also contributes to the large overall GSD.[…]

Conclusions:
[…]The observed rBC and EC mass concentrations correlated well with each other. However, the median of the observed rBC to EC mass ratios varied from 0.53 to 1.29 from campaign to campaign. Potential reasons for discrepancies are as follows:[…]

2) My second concern is about the way the role of tar brown carbon is addressed in the revision (see Section 2.2.1). The reference and the quoted statement "...its absorbance decreases strongly from the blue–UV region of the electromagnetic spectrum towards the red region (Karanasiou et al., 2015), thereby reducing the potential impact of brown carbon interference" is outdated given the fact the significant red absorption of tar brown carbon has been only recently discovered (see references in my previous comments). The addition of "per unit mass" does not resolve the issue since tar brown carbon is in the form of large tar balls of several hundred nm from biomass burning emission (which is the most significant single source of PM2.5 in Europe according to

several 14C studies), therefore their smaller mass-specific absorption may not necessarily mean that their overall contribution is not significant.

We modified Sect. 2.2.1 in order to put more emphasis on how soluble brown carbon and tar brown carbon can interfere with EC measurements. The modified text reads:

[…]Moreover, soluble brown carbon on filters can affect the laser correction if it was evolving during the OC steps, thereby causing a positive EC artefact. However, soluble brown carbon absorbs much less per unit mass than EC at the red wavelength ($\lambda$ = 635 nm) of the laser used in the thermal–optical instruments, since its absorbance decreases strongly from the blue–UV region of the electromagnetic spectrum towards the red region (Karanasiou et al., 2015). This reduces the potential interference of soluble brown carbon via the introduction of a bias in the optical charring correction. Recently, Massabò et al. (2019) developed a modified Sunset Lab Inc. EC/OC analyser to measure the brown carbon content in the sample by adding a second laser diode at $\lambda$ = 405 nm.
Tar brown carbon only evolves in the oxidizing step of TOA due to its refractoriness (Corbin et al., 2019). Therefore, it is assigned to EC independent of its light absorption properties. This is in contrast to LII, where tar brown carbon only gives marginal contribution to observed rBC mass (Sect. 2.3.3).[…]

We also added more discussion in Sect. 3.3.3 to address the hypothesis of tar brown carbon interference:

[…]Furthermore, tar brown carbon has been shown to be assigned to EC mass in TOA (Sect. 2.2.1), while it does not contribute to rBC mass in LII (Sect. 2.3.3). Such tar brown carbon interference would cause a negative relationship of data points as presented in Fig. S4, which was not observed. Hence, the observations do not provide evidence of substantial fraction of tar brown carbon in total EC in daily averaged samples. We conclude that the variation of BC sources and carbonaceous aerosol composition, as implied by AAE variability, may contribute to variations in the discrepancy between $m_{EC}$ and $m_{rBC}$, while not being the main driver of it.[…]